# Topographic and vegetation controls of the spatial distribution of snow depth in agro-forested environments by UAV-lidar

Vasana Dharmadasa [1,2,3,5], Christophe Kinnard [1,2,3], and Michel Baraër [4,5]

[1]Department of Environmental Sciences, University of Québec at Trois-Rivières, QC G8Z 4M3, Canada
5  [2]Center for Northern Studies (CEN), Québec City, QC GV1 0A6, Canada
[3]Research Centre for Watershed-Aquatic Ecosystem Interactions (RIVE), University of Québec at Trois-Rivières, Trois-Rivières, QC G8Z 4M3, Canada
[4]Department of Construction Engineering, École de technologie supérieure, Montréal, QC H3C 1K3, Canada
[5]CentrEau, the Québec Water Management Research Centre, Québec City, QC GV1 0A6, Canada

10  *Correspondence to*: Vasana Dharmadasa (vasana.sandamali.dharmadasa@uqtr.ca)

**Abstract.** Accurate knowledge of snow depth distributions in forested regions is crucial for applications in hydrology and ecology. Understanding and assessing the effect of vegetation and topographic conditions on snow depth variability is useful for accurate prediction of snow depths. In this study, the spatial distribution of snow depth in two agro-forested sites and one coniferous site in eastern Canada was analyzed for topographic and vegetation effects on snow accumulation. Spatially 15  distributed snow depths were derived by Unmanned Aerial Vehicle Light Detection and Ranging (UAV-lidar) surveys conducted in 2019 and 2020. Distinct patterns of snow accumulation and erosion in open areas (fields) versus adjacent forested areas were observed in lidar-derived snow depth maps at all sites. Omnidirectional semi-variogram analysis of snow depths showed the existence of a scale break distance of less than 10 m in the forested area at all three sites, whereas open areas showed comparatively large scale break distances (i.e., 11–14 m). The effect of vegetation and topographic variables on the 20  spatial variability of snow depths at each site was investigated with random forest models. Results show that including wind-related forest edge proximity effects in agro-forested sites and incorporating canopy characteristics in the coniferous site increased the model prediction accuracy by more than 90 %. Hence the underlying topography and the wind-redistribution of snow along forest edges govern the snow depth variability at agro-forested sites, while forest structure variability dominates snow depth variability in the coniferous environment. These results highlight the importance of including and better 25  representing these processes in physically-based models for accurate estimates of snowpack dynamics. This study also demonstrates the usefulness of UAV-lidar to resolve and understand high-resolution snow depth heterogeneity in agro-forested environments and boreal forests.

## 1   Introduction

Knowledge of spring snowpack conditions is essential to accurately estimate water availability and flood peaks following the 30  onset of melt (Hopkinson et al., 2004). Many studies showed that addressing the spatial distribution of snow depth prior to melting is more important than spatial differences in melt behavior when estimating melt dynamics of the snowpack  (e.g.,

Schirmer and Lehning, 2011; Egli et al., 2012). Evaluating snowpack conditions in forested regions is particularly crucial as the forest cover significantly modifies snow accumulation and ablation processes due to canopy interception and changes energy balance processes within the canopy. These changes produce a marked effect on downstream hydrographs (Roth and

Nolin, 2017). In addition, forests can also influence differential snow accumulation by preferential deposition of wind-blown snow along the forest edges (Essery et al., 2009; Currier and Lundquist, 2018).

Spatial variability of the snow cover is mainly controlled by topography, vegetation type, and vegetation density (Golding and Swanson, 1986; Jost et al., 2007; Varhola et al., 2010a; Koutantou et al., 2022). With the advent of remote sensing techniques, airborne (piloted and unpiloted) laser (lidar: light detection and ranging) scanning techniques have been extensively used to

monitor snowpacks due to their strong penetration ability through the canopy to detect underlying snow cover/ground (Hopkinson et al., 2004; Morsdorf et al., 2006; Hopkinson et al., 2010; Deems et al., 2013; Harpold et al., 2014; Zheng et al., 2016; Currier and Lundquist, 2018; Zheng et al., 2018; Mazzotti et al., 2019; Harder et al., 2020; Jacobs et al., 2021). Lidar scanning also typically allows capturing high-resolution micro variability and allows producing high resolution (<10 m) snow depth/cover maps (e.g., Deems et al., 2013; Harder et al., 2020; Koutantou et al., 2021; Dharmadasa et al., 2022).

Snow spatial variability can occur on more than one scale due to different processes acting over multiple scales (Deems et al., 2006; Clark et al., 2011). Several studies emphasized a multiscale behavior of snow depths with two distinct regions (scales) separated by a scale break at a location varying from meters to tens of meters, with a more strongly spatially correlated snow depth structure before the scale break (Deems et al., 2006; Fassnacht and Deems, 2006; Trujillo et al., 2007; Deems et al., 2008; Trujillo et al., 2009; Mott et al., 2011; Schirmer and Lehning, 2011; Helfricht et al., 2014; Clemenzi et al., 2018;

Mendoza et al., 2020a; Mendoza et al., 2020b). In turn, this suggests the existence of different combinations of processes controlling the snow accumulation, and distribution over these two distinct scales. For instance, these studies emphasized that canopy interception causes a short scale break distance in forested areas (9–12 m) where the effect of wind redistribution is minimal (Deems et al., 2006; Trujillo et al., 2007). Comparatively longer distances (15–65 m) were reported in tundra regions and explained by the interaction of wind, vegetation, and terrain roughness (Trujillo et al., 2009), while a shorter (6 m) and

longer (20 m) distance in non-vegetated areas are explained by the interaction of the wind with terrain roughness in sheltered and exposed mountain slopes, respectively (Mott et al., 2011; Schirmer and Lehning, 2011). The estimation of this scale break location is important when choosing the horizontal resolution required for remotely sensed or in situ data collection efforts, and model scales in order to represent the snowpack variability at different scales.

In addition to the scaling properties of snow distribution, the relationship between snow depth, topography, and forest structure

is also an important aspect in understanding/assessing small-scale snow heterogeneity in forested environments. The need to quantify these complex relationships has inspired the development of numerous empirical models (e.g., Anderton et al., 2004; Winkler et al., 2005; Grünewald et al., 2013) and process-based models (e.g., Hedstrom and Pomeroy, 1998; Liston and Elder, 2006; Mazzotti et al., 2020a; Mazzotti et al., 2020b). While process-based models are applicable to a wide range of conditions, they do require an extensive amount of input data. Contrarily, empirical models are useful in establishing a general relationship

between the variables and provide a first-order estimate of their effects on snow processes. However, they do not explicitly

account for governing processes, and thus may not make accurate predictions under specific conditions (Varhola et al., 2010a). Nevertheless, the use and effectiveness of empirical models like multiple linear regressions (MLR) (Jost et al., 2007; Lehning et al., 2011; Grünewald et al., 2013; Revuelto et al., 2014; Zheng et al., 2016; Zheng et al., 2018) and binary regression trees (BRT) (Elder et al., 1995; Elder et al., 1998; Winstral et al., 2002; Anderton et al., 2004; Molotch et al., 2005; Baños et al.,

2011; Revuelto et al., 2014) to relate snow depth/SWE patterns with terrain and land cover predictors is well documented. Compared to linear methods, tree-based methods have the ability to describe more complex and nonlinear relationships between snow depth and landscape variables (Erxleben et al., 2002; Veatch et al., 2009; Bair et al., 2018). In recent years, the random forest (RF) model, an ensemble machine learning algorithm that combines several randomized decision trees and aggregates their predictions, started gaining popularity in water science and hydrological applications (Tyralis et al., 2019).

The use of the ensemble bagging approach in RF models reduces overfitting, which is a well-known issue with traditional decision trees, and provides more accurate and unbiased error estimates (Breiman, 2001). As yet, there is only a handful of studies that used RF models to estimate snow depths/SWE (Bair et al., 2018; Yang et al., 2020) other than those that used RF algorithm to quantify the relative importance of predictor variables (Zheng et al., 2016) or to predict spatially distributed lidar vertical errors (Tinkham et al., 2014).

To our knowledge, to date, there are only a few previous studies that estimated snow depths by unpiloted aerial vehicle (UAV) based lidar (Harder et al., 2020; Cho et al., 2021; Jacobs et al., 2021; Koutantou et al., 2021; Dharmadasa et al., 2022). None of them explicitly examined how terrain and vegetation characteristics influence snow heterogeneity in different landscapes. From previous studies, Koutantou et al. (2022) successfully used UAV-lidar data on two opposing slopes with a heterogeneous forest cover at a high spatio-temporal scale to show the effect of canopy structure and solar radiation on snow dynamics,

excluding the effect of microtopography. The main objective of this paper is to study the small-scale spatial variability of snow depth by UAV-lidar and investigate the terrain (including the effect of microtopography) and vegetation controls on this snow depth heterogeneity in an agro-forested and a boreal landscape. The study sites are based in southern Québec, Canada, where forests intertwined with mosaics of open agricultural fields in low-lying lands (agro-forested landscapes) play a significant role in altering the spatial distribution of the snow cover (Aygün et al., 2020). Much uncertainty still exists about the micro

and meso scale spatial variability of snow cover and associated hydrological processes in these landscapes, partly due to lack of detailed and simultaneous micrometeorological and snowpack observations (Brown, 2010; Sena et al., 2017; Valence et al., 2022). To our knowledge, there has been no application of UAV laser scanning to investigate the small-scale snow cover heterogeneity in this type of landscape. This study will specifically explore: (1) how the snow accumulation and its scaling characteristics vary between and within forested and open environments, and (2) the relationship between snow depth,

topography, and forest structure in different sites. Motivated by previous works (Currier and Lundquist, 2018; Mazzotti et al., 2019), we specifically investigate how the forest edges modulates the accumulation patterns in agro-forested environments. Given the relatively flat topography in these environments, we postulate that preferential accumulation along forest edges may represent a significant factor of spatial variability in snow depth.

## 2 Data and methods

### 2.1 Study sites

Small-scale snow depth heterogeneity was investigated at three selected sites that represent the typical landscape in southern Québec (Fig. 1). Of the three sites, Sainte-Marthe and Saint-Maurice are agro-forested sites located in the St. Lawrence River lowlands. Irrigation canals and streams flowing through the open agricultural areas are very common in these agro-forested landscapes. The main crop type in the agricultural areas is soya. The forested area in Sainte-Marthe consists of a dense deciduous forest with sugar maple (*Acer saccharum*), red maple (*Acer rubrum*), and a small conifer plantation to the southwest. Saint-Maurice has a high to moderate dense mixed forest with poplar (*Populus x canadensis*), red maple, white pine (*Pinus strobus*), and balsam fir (*Abies balsamea*) being the dominant tree species. Forêt Montmorency (hereafter Montmorency) is a dense boreal forest with balsam fir, black spruce (*Picea mariana*), and white spruce (*Picea glauca*) tree species farther north on the Canadian Shield. Forest gaps associated with clear-cutting and regeneration practices are common in this area. Adjacent to the forest is an open area hosting the NEIGE-FM snow research station, which hosts a variety of precipitation gauges and snowpack measuring sensors, and is part of the World Meteorological Organization's (WMO) station network (Royer et al., 2021). Table 1 summarizes the physiographic and climatic conditions at each site. Land use information presented in Fig.1 was obtained from the Québec Ministry of Forests, Wildlife, and Parks (MFFP). For the interpretation purpose, open agricultural areas in Sainte-Marthe and Saint-Maurice and the small open area in Montmorency (NEIGE-FM site) are referred to as "field" herein.

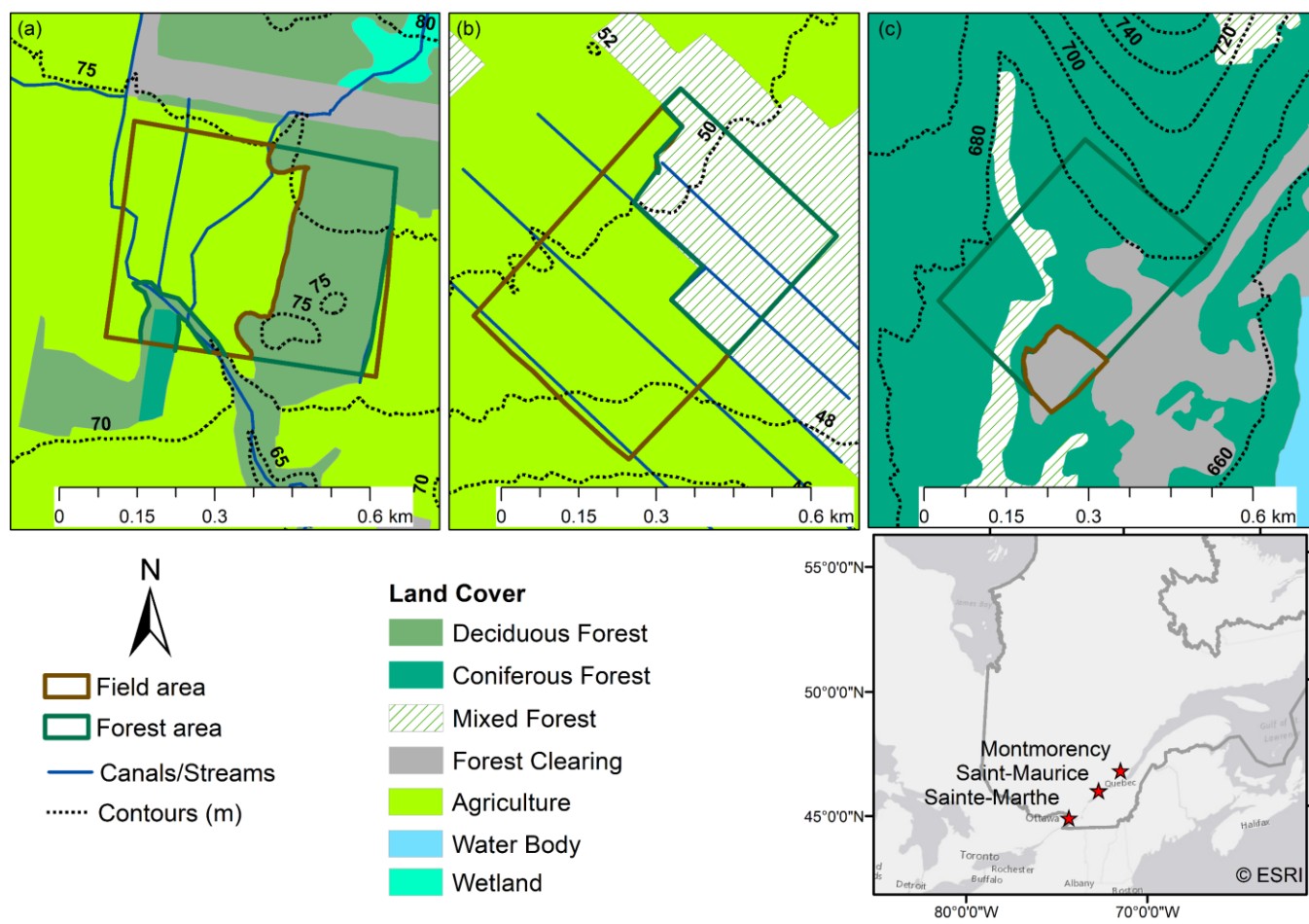

**Figure 1.** Overview of the study sites with lidar survey extents. Field and forest areas within each lidar extent are delineated with brown and green colors, respectively. **(a)** Sainte-Marthe, **(b)** Saint-Maurice and **(c)** Montmorency. Contour intervals intentionally differ between sites for better readability. (Adapted from Dharmadasa et al. (2022))

**Table 1.** Site characteristics and lidar data collection information (Adapted from Dharmadasa et al. (2022))

|  | Sainte-Marthe | Saint-Maurice | Montmorency |
|---|---|---|---|
| Elevation range, m | 70–78 | 46–50 | 670–700 |
| MAAT, °C | 6.0 | 4.7 | 0.5 |
| Total precipitation, mm/yr | 1000 | 1063 | 1600 |
| Snowfall/Total Precipitation, % | 15 | 16 | 40 |
| Winter season | November–March | November–March | October–April |
| Lidar survey extent, km$^2$ | 0.22 | 0.25 | 0.12 |
| Forest area/Total area, % | 40 | 40 | 92 |
| Forest type | Deciduous | Mixed | Boreal |

| Mean canopy density, % | >80 | 60–80 | 60–80 |
|---|---|---|---|
| Snow-on flight date | 12 March 2020 | 11 March 2020 | 29 March 2019 |
| Snow-off flight date | 11 May 2020 | 02 May 2020 | 13 June 2019 |

MAAT= mean annual air temperature. Climatic data presented here were based on the climate averages (1981–2010) at the nearest Environment and Climate Change Canada (2021b) meteorological stations to the sites (Station climate ID 7016470, 7017585, and 7042388 for Sainte-Marthe, Saint-Maurice and Montmorency). None of the snow-on flights were conducted right after a storm.

Although the lidar data acquisition years are different between agro-forested sites and boreal forest due to logistical reasons, the study years are representative of the long-term climatological conditions at the sites (Supplement Fig. S1), and hence allowed us for inter-site comparison of snow depths.

## 2.2 Data processing

All lidar surveys were performed with a GeoMMS system mounted onto a DJI M600 Pro UAV platform. The GeoMMS system
is comprised of a Velodyne VLP-16 lidar sensor, a real-time dual-antenna global navigation satellite system (GNSS) aided inertial navigation system (INS) for precise heading, and a tactical MG364 inertial measurement unit (IMU). The nominal accuracy of the point cloud provided by GeoMMS is ±5 cm (RMS, root mean square) (Geodetics, 2018) whereas the nominal uncorrelated relative error of two lidar point clouds is approximately ±7 cm ($\sqrt{5^2 + 5^2}$). Flight paths for the surveys were prepared in UgCS flight control software (Sph-Engineering, 2019) and the flight parameters were optimized to reduce overall
INS errors and maximize the mapping efficiency in the forested areas. Table 2 outlines the flight parameters and equipment settings used in surveys.

Raw lidar data sets collected from the flights were post-processed in Geodetics LiDARTool (Geodetics, 2019) with post-processing kinematic (PPK) correction. The PPK option regenerated a significantly more accurate trajectory file by combining the onboard GNSS data with GNSS base station data. Then, this post-processed trajectory file was merged with the raw laser
data to produce a geo-referenced x,y,z point cloud. Noise removal was applied next. We also employed a trial-and-error, manual boresight calibration method to correct for boresight errors in the data, as recommended by the manufacturer (Geodetics, 2019). The final post-processed point clouds have a vertical absolute accuracy range of 3–6 cm and a relative accuracy range of 4–6 cm (Dharmadasa et al., 2022).

To classify the bare surface points, we used the multiscale curvature algorithm (Evans and Hudak, 2007) implemented in the
commercial Global Mapper software (Blue Marble Geographics, 2020). Parameters of the algorithm were adjusted according to the vertical spread of the flight strips over open terrain, the local slope of the terrain and canals/streams, and the presence/absence of buildings. The reader is referred to Dharmadasa et al. (2022) for a comprehensive overview of the UAV-lidar system and post-processing of raw data.


**Table 2.** Flight parameters and equipment settings

| Flight parameters | | Equipment settings | |
|---|---|---|---|
| Flying speed | 3 m s$^{-1}$ | Wavelength | 905 nm |
| Flight altitude | 40 m AGL | Laser pulse repetition rate | 18.08 kHz |
| Field of view (horizontal) | 145° | Field of view (vertical) | ±15° |
| Distance between parallel flight lines | 64 m | Laser RPM | 1200 |
| Ground overlap | 20 % | Return type | Dual |
| Point density | 603 points m$^{-2}$ | | |

### 2.2.1    Snow depth maps

Snow depth maps were obtained by differencing winter (snow-on) and summer (snow-off) digital elevation models (DEMs) generated from bare surface points at each site. Bare surface points were aggregated to a grid resolution of 1.4 m using the binning method in Global Mapper (Blue Marble Geographics, 2020). This grid resolution was selected based on the manual snow depth sampling strategy used by Dharmadasa et al. (2022) to validate the snow depth maps and aimed to minimize the effect of positional errors of the manual measurements made with GNSS. The manual sampling strategy consisted of five snow depth measurements taken at each sampling location in a diagonal cross shape at 1 m apart, and the average of these five measurements represents a 1.4x1.4 m ($\sqrt{1^2 + 1^2}$) grid cell. As final filtering, spurious negative snow depths were set to zero, as they are physically inconsistent and need to be filtered (Hopkinson et al., 2012). Negative snow depths accounted for a very small portion of the total area (<0.1 %) sampled and had a negligible effect on the statistics derived from the snow depth maps. The validation of UAV-lidar snow depths with manual measurements showed a RMSE of 0.079–0.160 m in the deciduous forested environment, and 0.096–0.190 m in the coniferous forested environment (Dharmadasa et al., 2022), which is comparable to previous efforts with UAV-lidar (Harder et al., 2016; Jacobs et al., 2021) and airborne lidar (Harpold et al., 2014; Painter et al., 2016). More details about the snow depth validation can be found in Dharmadasa et al. (2022).

### 2.2.2    Terrain metrics

To typify the terrain characteristics, we derived four variables from the summer DEM, i.e., elevation (*Elevation*), slope (*Slope*), aspect (*Aspect*), and topographic wind sheltering index (*TWSI*) at 1.4 m resolution (Supplement Fig. S2–S4). Topographic variables other than elevation need to be considered when studying areas that encompass a small elevation range (Zheng et al., 2016), such as our sites. *Elevation* was obtained directly from the DEM, while *Slope* and *Aspect* were derived using ArcGIS 10.2 software. *Slope* was calculated as the first derivative of the DEM, while *Aspect* was derived in two orthogonal components, i.e., west-east (*Aspect_WE*) and south-north (*Aspect_SN*) exposures. *Aspect_WE* (west-negative, east-positive) and *Aspect_SN* (south-negative, north-positive) were calculated directly as the sine and cosine of the aspect, respectively. The *TWSI* was

produced using the RSAGA package in CRAN. This variable considers the sheltering effects of the local topography in the dominant wind direction. Several studies showed that *TWSI* is a good measure to characterize sheltering and exposure of the local terrain providing a reasonable representation of the local wind field and thus the redistribution of snow by wind (Winstral et al., 2002; Winstral and Marks, 2002; Plattner et al., 2004; Molotch et al., 2005). Negative *TWSI* values correspond to terrain exposure and positive values to sheltering from the wind. Dominant wind directions were extracted from hourly wind data for the study period considered (winter season in each study year as indicated in Table 1) at each site (Fig. 2). Wind data was collected from an automatic weather station located 1.4 km away from the Sainte-Marthe site and the closest Environment Canada wind measuring stations at the other sites. The closest station to Saint-Maurice (climate ID 7018561) was 19 km away from the site and 0.25 km away from the Montmorency site (climate ID 7042395) (ECCC, 2021a).

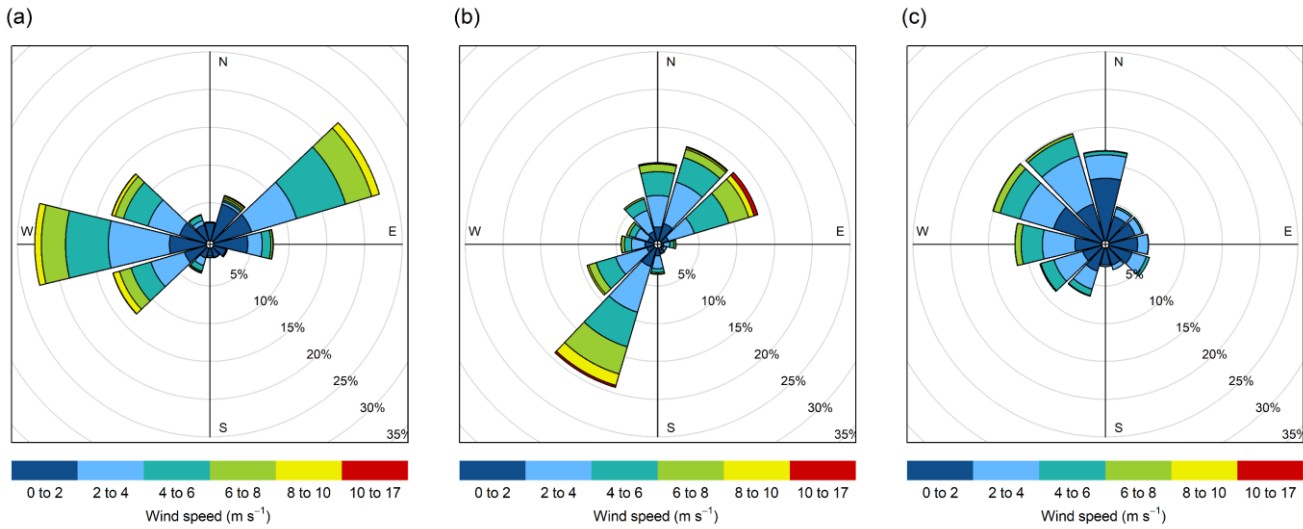

**Figure 2.** Winter period wind rose plots of the sites. **(a)** Sainte-Marthe, **(b)** Saint-Maurice and **(c)** Montmorency

### 2.2.3  Vegetation descriptors

Vegetation-related variables were rasterized from the classified winter point cloud in LiDAR360 (Greenvalley-International, 2020). The forestry module of LiDAR360 contains tools that allow users to calculate essential forest metrics and accurately extract individual tree parameters like crown diameter, crown area, and tree diameter by breast height from airborne lidar data. In this study, the leaf area index (*LAI*), canopy cover (*CC*), and gap fraction (*GF*) were estimated at 1.4 m resolution for the forest cover higher than 2 m (Supplement Fig. S2–S4). A 2 m height threshold was selected as canopies >2 m has been shown to have a strong influence on snow accumulation (Varhola et al., 2010b; Zheng et al., 2016; Zheng et al., 2019). The function used to calculate *LAI* is based on the Beer-Lambert law (Richardson et al., 2009). The estimated *LAI* is contingent on the average scan angle, *GF,* and extinction coefficient. *GF*, the amount of open area within the canopy which is not blocked by branches or foliage, is calculated as the total number of ground points to the total number of lidar points within a grid cell. *CC*, which is defined as the percentage of vertical projection of forest canopy to the forest land area (Jennings et al., 1999), is

calculated as the total number of vegetation returns to total returns (Morsdorf et al., 2006), ($CC = 1 - GF$). Refer to Richardson et al. (2009) and Morsdorf et al. (2006) for the equations used by LiDAR360 to estimate the forest metrics. In addition, canopy height ($CH$) was derived by subtracting the DEM from the digital surface model (DSM).

### 2.2.4 Site variable

A binary variable, *Site* representing forested (1), and field (0) pixels was derived to investigate systematic effects, if any, of land cover that was not captured by vegetation or terrain metrics (Supplement Fig. S2–S4). This variable was derived by manually mapping field and forested area boundaries at each site in ArcGIS 10.2 software. After delineating forest and field boundaries, the area inside the forest boundary was assigned a value of 1, and the area inside the field boundary was assigned a value of 0.

### 2.2.5 Forest edge descriptors

We investigated forest edge effects on snow accumulation using an approach inspired from Currier and Lundquist (2018) and Mazzotti et al. (2019) using Matlab software. Analogous to their analyses, we added directionality to forest edges to examine if preferential snow accumulation occurred windward or leeward of forest edges due to snow redistribution by wind or reduced ablation due to shading from the forest. Pixels were first classified as north-facing ($NFE$) when they were within a maximum search distance $d_{max}$ northward of the forest edge. Forest edges (the boundary between field and forest areas) were extracted from the *Site* variable. Based on previous results by Currier and Lundquist (2018), $d_{max}$ was set to 2H, where H is the typical tree height derived from the canopy height model at each site. The 2H distance reflects the typical shading of the ground by the canopy. H is 15 m in Sainte-Marthe, 20 m in Saint-Maurice, and 12 m in Montmorency. A tolerance of ±45° was used for the search direction for $NFE$. Pixels were further classified as windward ($WFE$) and leeward ($LFE$) when they were within a maximum search distance of the forest edge in the dominant wind direction. A range of search directions was used to constrain the dominant wind directions at each site, based on wind roses (Fig. 2). Two dominant wind cones, 270±15°, and 50±15° were used in Sainte-Marthe, and one dominant wind cone in Saint-Maurice (210±15°) and Montmorency (310±15°). $d_{max}$ was initially varied between 6–10H for pixels in open terrain based on Currier and Lundquist (2018), which represents the typical length scale of preferential snow accumulation at the forest edge. After a few trials, a final value of 10H was retained, which showed the highest correlation with snow depth. Moreover, the 10H distance at each site (150 m, 200 m, and 120 m in Sainte-Marthe, Saint-Maurice, and Montmorency respectively) encompassed the preferential snow accumulation seen along the forest edges on the lidar-derived snow depth maps. A maximum search distance of 1H was used for pixels within the forest in order to detect if preferential accumulation from blowing snow penetrated the forest. This value was chosen based on visual observations in the field, which suggested limited penetration of blowing snow inside the forest. Figure 3 shows a schematic illustration of the forest edge parameters described.

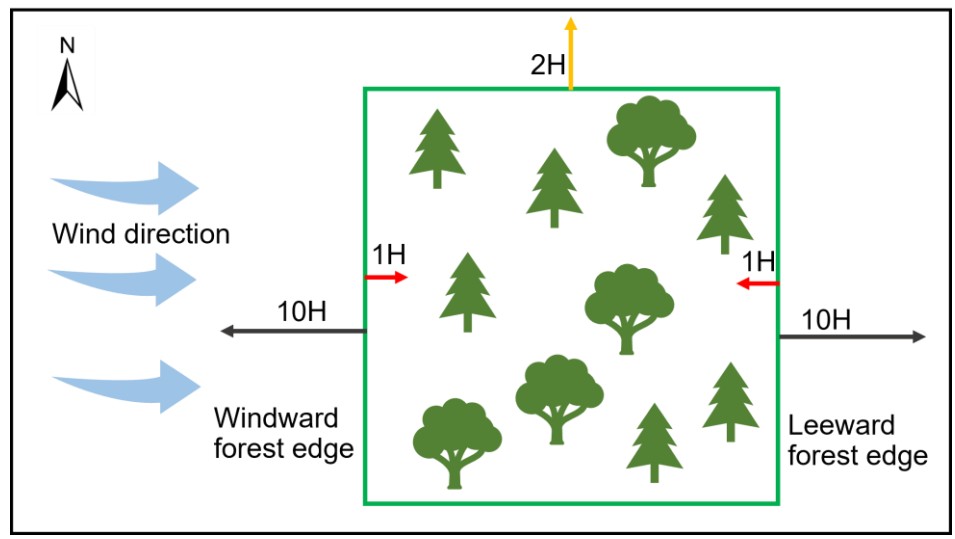

**Figure 3.** Graphical illustration of forest edges and respective maximum search distances, $d_{max}$. 10H indicates the maximum search distance in the open field from the forest edge in windward and leeward direction, 1H indicates the maximum search distance in the forest from the forest edge in the windward and leeward direction, and 2H indicates the maximum search distance northward of the forest edge, for shading effects.

A new index of proximity to the forest edge, *FE*, was calculated by scaling the distance between each pixel and the forest edge (*d*) by the maximum search distance, $d_{max}$:

$$FE = \frac{d_{max}-d}{d_{max}}$$  (1)

*FE* (either *NFE*, *WFE,* or *LFE*, depending on the initial classification) is equal to one when a pixel is situated on the forest edge and equal to zero when it is located at, or beyond the maximum search distance $d_{max}$. The novelty of this approach is to derive a continuous predictor of forest edge proximity, as opposed to the simpler binary classification introduced by Currier and Lundquist (2018). Maps of the forest edge descriptors for each site can be found in supplement Fig. S2–S4.

## 2.3  Data analysis

Data analysis was primarily focused on assessing the small-scale snow depth heterogeneity at the selected sites. Lidar-derived snow depth data were analyzed for inter (agro-forested versus coniferous) and intra (field versus forest) site variability. First, the scale dependence of snow depth variability was explored using semi-variogram analysis. Then, the site-specific topographic and vegetation control on the snow depth spatial heterogeneity was examined with RF regression models. All the statistical analyses were performed in R software.

### 2.3.1    Spatial correlation analysis

To analyze the small-scale spatial variability of the snow depth map in each study site, omnidirectional semi-variograms were used. Semi-variogram analysis allows constraining the dominant scales of snow depth variability and to compare them between

land cover types and sites. Canals/streams were discarded from the snow depth maps for this analysis to ensure stationarity of the surface. i.e., snow depths in canals/streams would have a unidirectional spatial correlation which could alter the relationship of the overall terrain by introducing biases. In addition, omnidirectional semi-variograms of snow depth were compared with those obtained from bare earth topography and topography+vegetation (DSM) surfaces to investigate the influence of topography and vegetation interactions on snow depth. Moreover, directional semi-variograms of snow depth were also computed to establish possible influences of dominant wind directions on snow depth variability at each site.

The semi-variogram $\gamma(r)$ is expressed as:

$$\gamma(r_k) = \frac{1}{2N(r_k)} \sum_{(i,j) \in N(r_k)} \{z_i - z_j\}^2 \tag{2}$$

Where $r$ is the lag distance of bin $k$, $N(r_k)$ is the total number of pairs of points in the $k^{th}$ bin and $z_i$ and $z_j$ are the snow depth values at two different point locations $i$ and $j$ (Webster and Oliver, 2007).

Half of the maximum point pairs distance (Sun et al., 2006) was taken as the maximum lag distance for the semi-variogram calculations with 50 log-width bins. Log-width distance bins provide equal bin widths when semi-variograms are transformed to log-log scale, and help resolve the semi-variogram at short length scales by allowing greater bin density at shorter lag distance compared to linear-width bins (Deems et al., 2006).

In the case of scale invariance, the semi-variogram can be described by a power law:

$$\gamma(r) = ar^b \tag{3}$$

Where $a$ and $b$ are coefficients selected to minimize the squared residuals.

To identify scale breaks in semi-variograms, the following steps were implemented following a similar approach suggested by Mendoza et al. (2020a).

- First, a change point analysis was conducted on the semi-variograms in log-log space using the ecp package in R (James and Matteson, 2014) to identify possible break points, which allows delineating sections of the semi-variogram with similar trends.

- Then, linear least square regression models were fitted in log-log space for each cluster of points identified in step 1.

- Finally, we checked whether the changes in the slopes of the log-log linear models were larger than 20 % and that the 95 % confidence limits of the slopes did not overlap, and verified that the $R^2$ was greater than 0.9. If all these conditions were fulfilled, the existence of a scale break was confirmed.

### 2.3.2 Random forest model

To investigate the effect of vegetation and topographic variables on the spatial variability of snow depth, we applied RF regression models on the rasters derived from lidar data. Data were not separated into training and test sets so that we would not create an artificial bias by data splitting. i.e., all data at each site were used in the RF analysis. Generally, in a RF model, two-thirds of the sample data (in-bag) are used to train the model, while the remaining one-third (out-of-bag, OOB) is used to

estimate how well the trained model performs. This in-bag and OOB sampling procedure is akin to the much used k-fold cross-validation approach (Probst and Boulesteix, 2017; Tyralis et al., 2019). As such, model performance statistics (mean square error, MSE and variance explained) are derived from the OOB predictions. The RF algorithm also calculates the predictor importance (importance of a variable), by estimating how much the prediction error increases when OOB data for the respective variable is permuted while all others are left unchanged (Liaw and Wiener, 2002).

The RF analyses were conducted in R with grid resolutions of 1.4 m at all sites. As a precautionary measure, we excluded collinear variables prior to building the RF models using the variance inflation factor (VIF) function in R. This was done mainly because our objective was to investigate the relative contribution of different variables to snow depth variability in forest versus the field, rather than deriving a model with maximum predictive capacity. While RF can handle collinearity in a predictive mode, collinearity makes it difficult to separately evaluate the predictive power (variable importance) of the

predictors (Bair et al., 2018). The number of trees in the ensemble (ntree) and the number of variables at each node (mtry) were tuned before training each RF model. RF model results were first examined for the relative importance of predictor variables (variable importance), which has proven to be useful for evaluating the relative contribution of input variables (Tyralis et al., 2019). Then the partial relationships of the variables with the snow depth were examined and presented. Partial dependence functions are typically used to help interpret models produced by machine learning models such as RF (Jerome,

2001). It is a risk-adjusted alternative to variable dependence. Each partial plot presented here was generated by integrating out the effects of all variables beside the covariate of interest. Partial dependence data in each plot were constructed by selecting points evenly spaced along the distribution of the variable of interest. This subsampling helps to cut down computational time substantially. We used the default subsampling of 51 points in our analysis. The performance of RF models in terms of OOB statistics was compared between the different land cover types and sites and presented next.

Additionally, we discuss RF model performances compared to traditional MLR models, as well as the relationships between snow depth and physiographic variables derived from RF models at 1.4 m resolution (sub-canopy resolution) versus single-tree scale at each site. Single-tree scale, as the name implies, is selected as the grid size that encompasses a single tree. This differed between the sites and was estimated in LiDAR360. We used the point cloud segmentation algorithm developed by Li et al. (2012) in LiDAR360 to segment individual trees and obtain their attributes such as tree location, tree height, crown

diameter, and crown area. Then, the maximum crown diameter of the segmented trees in each site was selected as the single-tree scale grid resolution. Single-tree scale resolutions obtained using this method were 20, 15, and 10 m in Sainte-Marthe, Saint-Maurice, and Montmorency, respectively.

## 3    Results

### 3.1    General snow accumulation patterns

Figure 4 depicts the snow depth maps derived from UAV-lidar data at the study sites. Montmorency shows the highest overall snow accumulation with a maximum of 3.6 m. Higher snow accumulation in canals/streams (area 1 in Fig. 4a, b) and along

the forest edge (area 2 in Fig. 4a, b) is evident in Sainte-Marthe and Saint-Maurice, whereas in Montmorency, forest gaps (area 4 in Fig. 4c) seem to accumulate more snow. The highest snow depth in Montmorency corresponds to localized, artificial snow piles adjacent to the main road as observed during the field campaign (area 5 in Fig. 4c). Concentric snow accumulation

patterns around the double fence precipitation gauges are also noticeable in Montmorency snow depth map (area 6 in Fig. 4c). Compared to the other two sites, the Montmorency snow depth map comprises more data gaps in the forested area. Paved roads in Sainte-Marthe (area 3 in Fig. 4a) and Montmorency (area 3 in Fig. 4c) and the area surrounding the small house (area 7 in Fig. 4a) in the forest at Sainte-Marthe appear snow-free due to the snow clearing operations, as confirmed in field campaigns. Snow clearing in the proximity of the house in Sainte-Marthe accounts for a significant portion of zero and/or

small snow depths (Fig. 4d) and biases the mean snow depth in the forest. When this portion is discarded, the mean snow depth in the forest increases from 0.250 to 0.275 m. In Sainte-Marthe, the mean snow depth in the field area is higher than that in the adjacent forested area (Fig. 4d), whereas, at the other two sites, mean snow depths in the field and forest are similar considering the measurement error of the lidar system (Fig. 4e, f). A nonparametric Wilcoxon rank-sum test (Wilcoxon, 1945) was applied to test whether snow depths within forested and field areas were statistically different from each other. To remove spatial

autocorrelation, snow depths were subsampled every 20 m (larger than the scale break distances found by semi-variogram analysis, Fig. 5). The results confirmed that snow depth in the Sainte-Marthe field was statistically greater than that in the forest and in the other two sites differences were not statistically significant.

Although the maximum snow depth is higher in Sainte-Marthe (1.8 m) compared to Saint-Maurice (1.6 m), snow depths in Sainte-Marthe are lower on average (mean forest = 0.250 m; mean field = 0.374 m) than in Saint-Maurice (mean forest = 0.591

m; mean field = 0.600 m). The snow depth is more variable in the forest (higher coefficient of variation, CV) than in the field in Sainte-Marthe and Montmorency, which is not the case in Saint-Maurice, where the coefficient of variation in the field is slightly larger than in the forest.

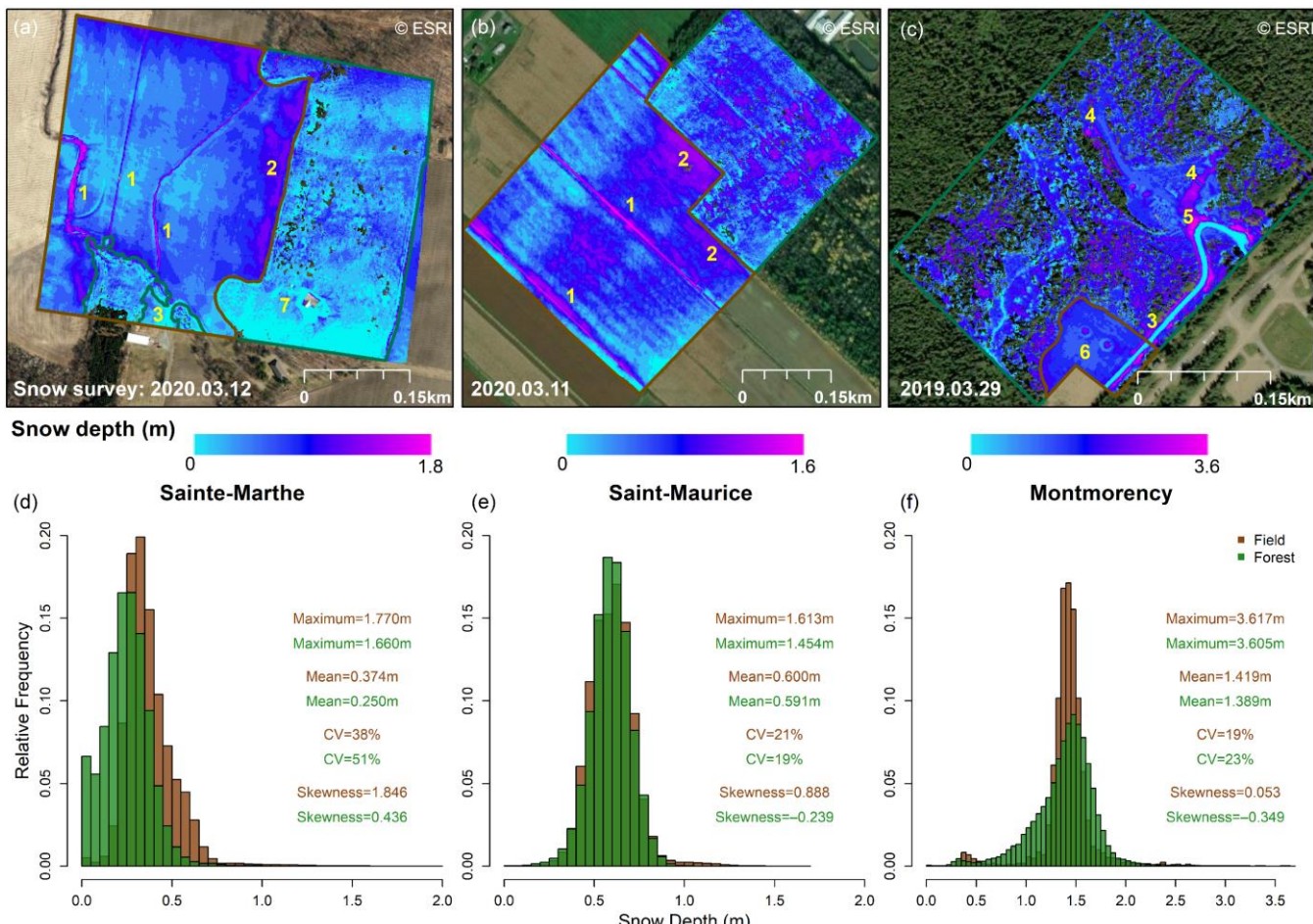

**Figure 4**. UAV-lidar derived snow depth maps (grid size 1.4 m) and histograms of snow depth distribution. **(a, d)** Sainte-Marthe map with snow surveying date and histogram; **(b, e)** Saint-Maurice map with snow surveying date and histogram; **(c, f)** Montmorency map with snow surveying date and histogram. Field and forest areas are demarcated with brown and green colors in snow depth maps respectively. Histograms are derived according to these boundaries. Features 1 to 7 are discussed in the text.

## 3.2 Spatial correlation analysis

Omnidirectional semi-variograms of snow depth, bare earth topography, and topography+vegetation surface at the study sites are shown on a log-log scale in Fig 5. Semi-variograms were discretely developed for field and forested areas to assess the effect of land cover on the snow depth variability. Overall, forested areas show more variable (higher semi-variance values) snow depths than field snow depths at all sites. Snow depths seem to be more variable in coniferous forests than in deciduous and mixed forests. Snow depth in forested areas at all three sites shows a typical multi-scaling behavior, where the semi-variance between neighboring snow depths increases rapidly up to a scale break located at distances less than 10 m (Fig. 5a, b and c), followed by a slower increase thereafter. Similarly, field snow depths exhibit multi-scaling behavior with comparatively larger scale break distances, with Montmorency showing two scale break distances (Fig. 5a, b and c). Topography+vegetation

surfaces show the highest semi-variance with scale break distances similar to forest snow depths (Fig. 5d, e and f). Sainte-Marthe bare earth topography does not exhibit a distinct scale break (Fig. 5d). In contrast, the bare earth topography at the other two sites shows multi-scaling behavior with scale break distances larger than 10 m (Fig. 5e, f).

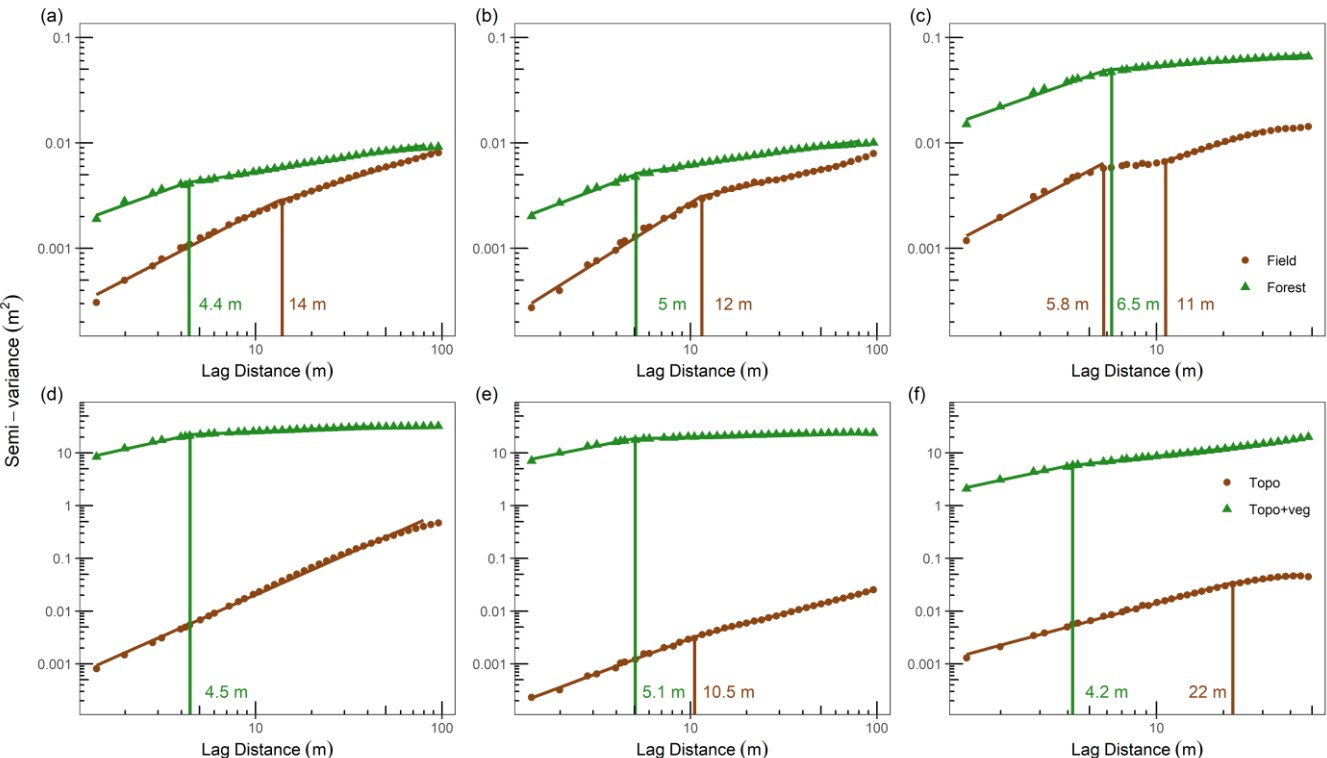

**Figure 5.** Omnidirectional semi-variogram for the field and forested areas for **(a)** Sainte-Marthe snow depth, **(b)** Saint-Maurice snow depth, **(c)** Montmorency snow depth, **(d)** Sainte-Marthe bare earth topography and topography+vegetation, **(e)** Saint-Maurice bare earth topography and topography+vegetation and **(f)** Montmorency bare earth topography and topography+vegetation. In the figure, Topo denotes bare earth topography and Topo+veg denotes topography+vegetation surface. Vertical lines indicate the dominant scale breaks, and trend lines represent significant ($p<0.05$) log-log linear models with $R^2 > 0.9$ (see methods).

Figure 6 shows directional semi-variograms of snow depth derived for field and forested areas at each site. Sainte-Marthe field snow depths show an isotropic behavior (Fig. 6a) whereas Sainte-Marthe forest shows an anisotropic behavior along the west-east direction (Fig. 6d). In contrast, both Saint-Maurice field and forest snow depths show distinct anisotropic behaviors. Saint-Maurice field snow depths show a narrow anisotropic pattern along northwest-southeast and a broad anisotropic pattern along southwest-northeast directions (Fig. 6b) whereas forest snow depths show an anisotropic pattern along southwest-northeast direction (Fig. 6e). Neither field nor forest snow depths in Montmorency show strong anisotropic behavior (Fig. 6c, f).

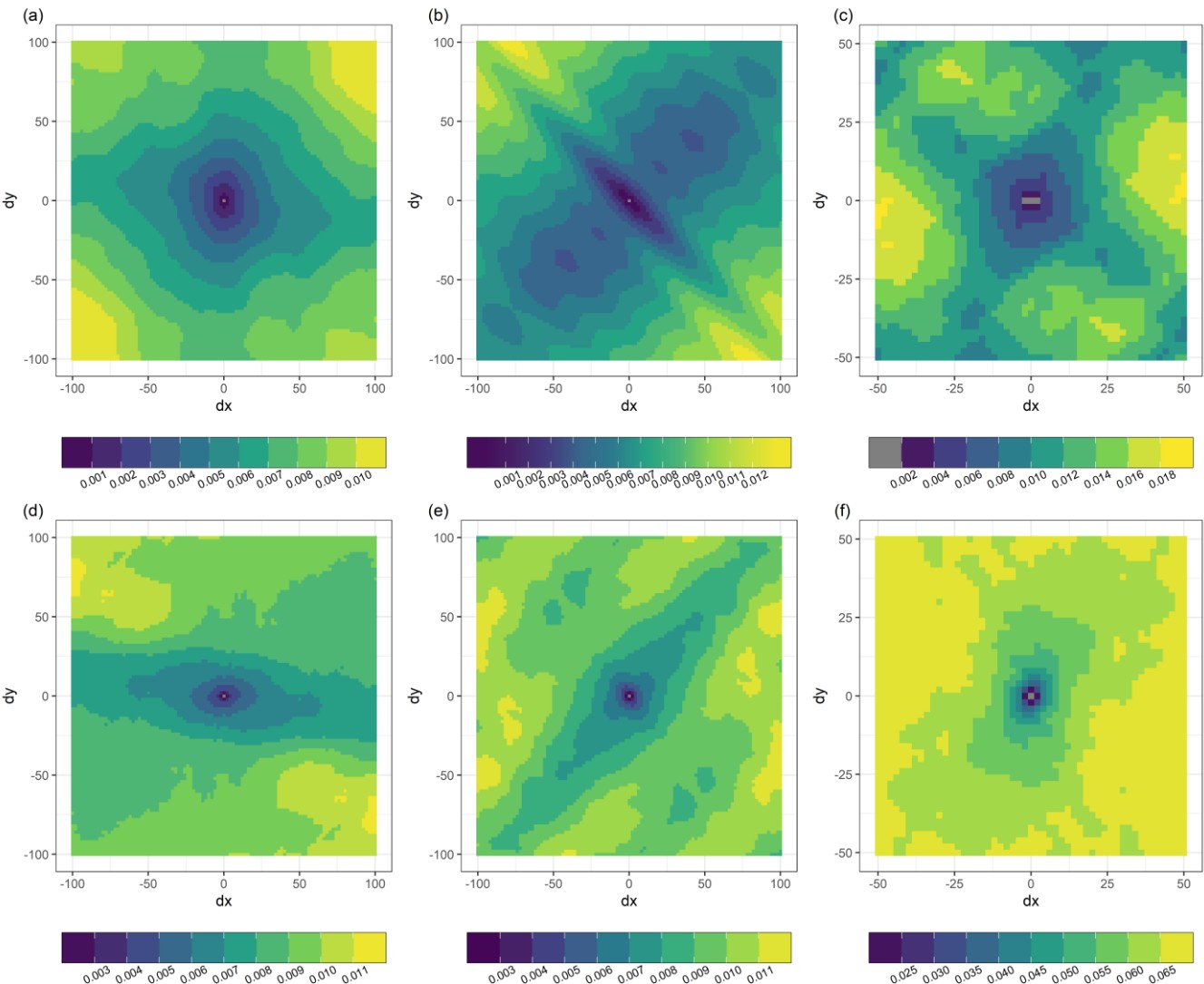

**Figure 6.** Directional semi-variogram of snow depth in **(a)** Sainte-Marthe field, **(b)** Saint-Maurice field, **(c)** Montmorency field, **(d)** Sainte-Marthe forest, **(e)** Saint-Maurice forest and **(f)** Montmorency forest

## 3.3 Random forest analysis

### 3.3.1 Potential predictors of RF model

*Elevation* was discarded from the analysis since the elevation range at all sites was too small (Table 1) to produce any meaningful local orographic effect on precipitation, or adiabatic effects on air temperature, e.g., Mazzotti et al. (2019), and could mask other local topographic effects on accumulation related to slope, aspect and terrain roughness (wind sheltering), due to collinearity. In addition, irrespective of the variable type, collinear variables were identified and discarded prior to

building the RF models at all sites. As such, the topographical variables *Slope*, *Aspect_WE*, *Aspect_SN*, and *TWSI* were used at all sites. However, the vegetation descriptors (*LAI*, *CC*, *GF*, and *CH*) were strongly intercorrelated (with correlation coefficient, r of 0.82–1.00) and hence could not be used together in a predictive model, at least not without compromising the interpretation of variable importance in the RF model. Therefore, *LAI* was selected as the most representative forest structure

indicator in the RF analysis as, it has been shown to be a strong predictor of snow accumulation in forests (Hedstrom and Pomeroy, 1998; Pomeroy et al., 1998; Broxton et al., 2015; Lendzioch et al., 2016). Moreover, a sensitivity analysis showed that the choice of forest structure descriptor has a negligible impact on the performance ($R^2$) of RF models (Supplement Table S1). The selection of the windward and leeward forest edge descriptors (*WFE* and *LFE*) was guided by the landscape setting at each site. In Sainte-Marthe, both *WFE* and *LFE* have large extents (Supplement Fig. S2) but are collinear due to the two

dominant and opposed wind directions. Including both variables in the RF model would thus compromise the interpretation of the variable importance. Hence, we opted to use the *WFE* only in the final RF analysis. In Saint-Maurice, *LFE* has only a few pixels (Supplement Fig. S3) and was hence omitted from the RF analysis. In Montmorency, *LFE* seemingly has more influence on snow depth variability with its larger extent than the *WFE* (as shown in Supplement Fig. S4). This is also more logical as the open areas in Montmorency constitute a large gap within an overall forested environment, so deposition is expected leeward

of the forest edge with little remobilization (erosion) within the gap. *NFE* was used at all sites to see the effect of forest edge shading on the snow depth variability.

### 3.3.2     Relative importance of topography and vegetation on snow depth variability

The relative importance of predictor variables in Fig. 7 summarizes the relative contribution of the different topographic, vegetation, and forest edge effects on snow depth spatial variability at each site. At the full domain (field+forest), windward

forest edge proximity (*WFE*) has the strongest influence on snow depth variability in both Sainte-Marthe (0.99) and Saint-Maurice (0.97), and the north-facing forest edge proximity (*NFE*) has the least influence (0.30 and 0.23). However, topographic wind sheltering (*TWSI*) exerts an equally strong impact on snow depth as *WFE* in Sainte-Marthe (0.99) compared to that in Saint-Maurice (0.70). In Montmorency, *LAI* and *NFE* have the highest (0.99) and least (0.07) impacts, respectively, on snow depth variability for the full domain. The importance of variables somewhat changes when forests and fields are modelled

independently, implying different dominant factors/processes acting in such environments. For instance, in Sainte-Marthe, the *TWSI* seems to be the dominant variable (0.74) for snow depth variability in the forest followed by *LAI* (0.36), *WFE* (0.36), and *Slope* (0.31). In Sainte-Marthe field, *WFE* (0.94), *TWSI* (0.87), and *Slope* (0.62) are the most important variables. *WFE* (0.33), *TWSI* (0.25), and *LAI* (0.21) have the highest influence on snow depth within the Saint-Maurice forest whereas in the adjacent field *WFE* (0.99), *TWSI* (0.64), and *Slope* (0.39) predominate. The importance of *LAI* (0.97), *TWSI* (0.41), and *Slope*

(0.25) is higher for snow depths within the coniferous forest with gaps in Montmorency, whereas the snow depths in the small field are mostly influenced by *LFE* (0.27), *TWSI* (0.23), and *Slope* (0.18).

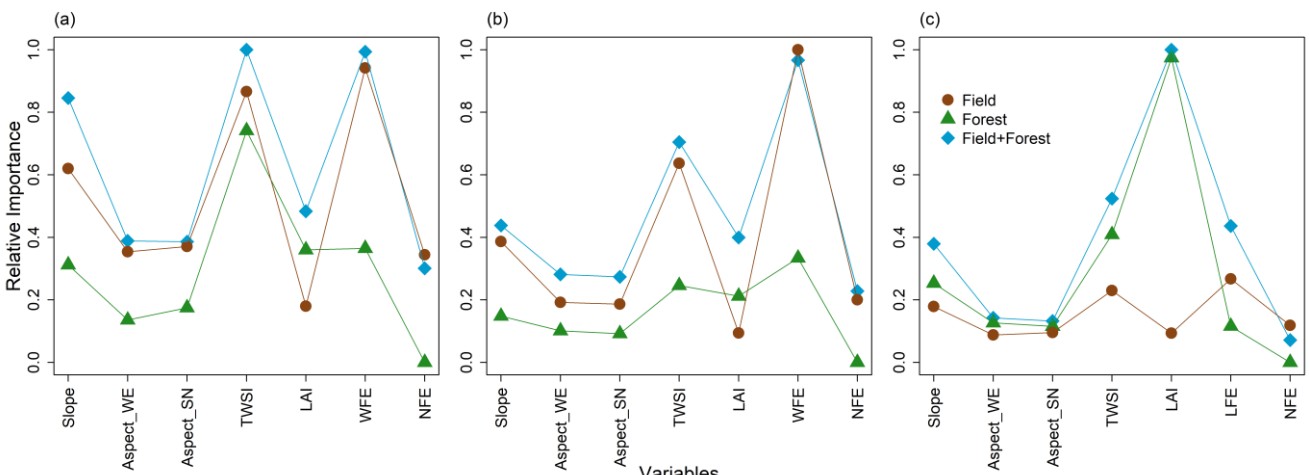

**Figure 7.** Relative importance of variables (scaled between 0 and 1) in predicting snow depths. **(a)** Sainte-Marthe, **(b)** Saint-Maurice and **(c)** Montmorency

### 3.3.3    Partial relationships of predictor variables with snow depth

As seen in Fig. 8, all variables exhibit mostly nonlinear relationships with snow depth across all sites. Spearman rank correlation coefficients ($\rho$) were used to quantify the strength of the partial relationships and reported in the graphs. A positive $\rho$ indicates an increasing monotonic trend and a negative $\rho$ indicates a decreasing one. Note that the positive *LAI* values in field areas correspond to a few isolated *LAI* pixels along the forest edges, the boundary between field and forest. In general, at all sites and despite the magnitude of the correlation, the two slope aspect variables (*Aspect_WE* and *Aspest_SN*) as well as forest shading represented by the north-facing forest edge proximity (*NFE*) have the least effect on snow depth change (i.e., a relatively flat partial relationship on Fig. 8). Moreover, all the relationships between landscape descriptors and snow depth for the overall domain in Montmorency (field+forest, blue curves on Fig. 8c), except *NFE*, are governed by the respective variable behavior in the forest, probably due to the large extent of forest at this site.

With regards to topographical control, all sites show increasing snow depths with increasing slopes in the field, forest, and field+forest (positive $\rho$ values in Fig. 8a, b, and c). The general relationship of snow depth with *TWSI* suggests that increased topographic sheltering from the wind (increasing *TWSI* values), leads to enhanced snow accumulation. At the two agro-forested sites (Fig. 8a, b), the greatest contribution to the overall field+forest *TWSI*-snow depth relation comes from field snow depths. As for the influence of vegetation, there is a decrease in snow depths in response to increasing *LAI* at all sites, although the relation is comparatively weak ($\rho$ = –0.65) in the Sainte-Marthe forest. Snow depth at the two agro-forested sites shows a general increase in response to increasing distance towards the windward forest edge (*WFE*), except within the Saint-Maurice forest. An increase of snow depth with *WFE* in Sainte-Marthe forest indicates more snow at the edge and decreasing inward the forest, which reflects blowing snow penetration from the field inside the forest. The increase in snow depth with *WFE* within the Sainte-Maurice forest for *WFE* > ~0.8 could also reflect the limited penetration of blowing snow from the field

inside the forest. In Montmorency, the field snow depth shows a non-linear relation with *LFE*, probably due to the influence of instrumentation while forest snow depths show a decrease in accumulation inward the forest.

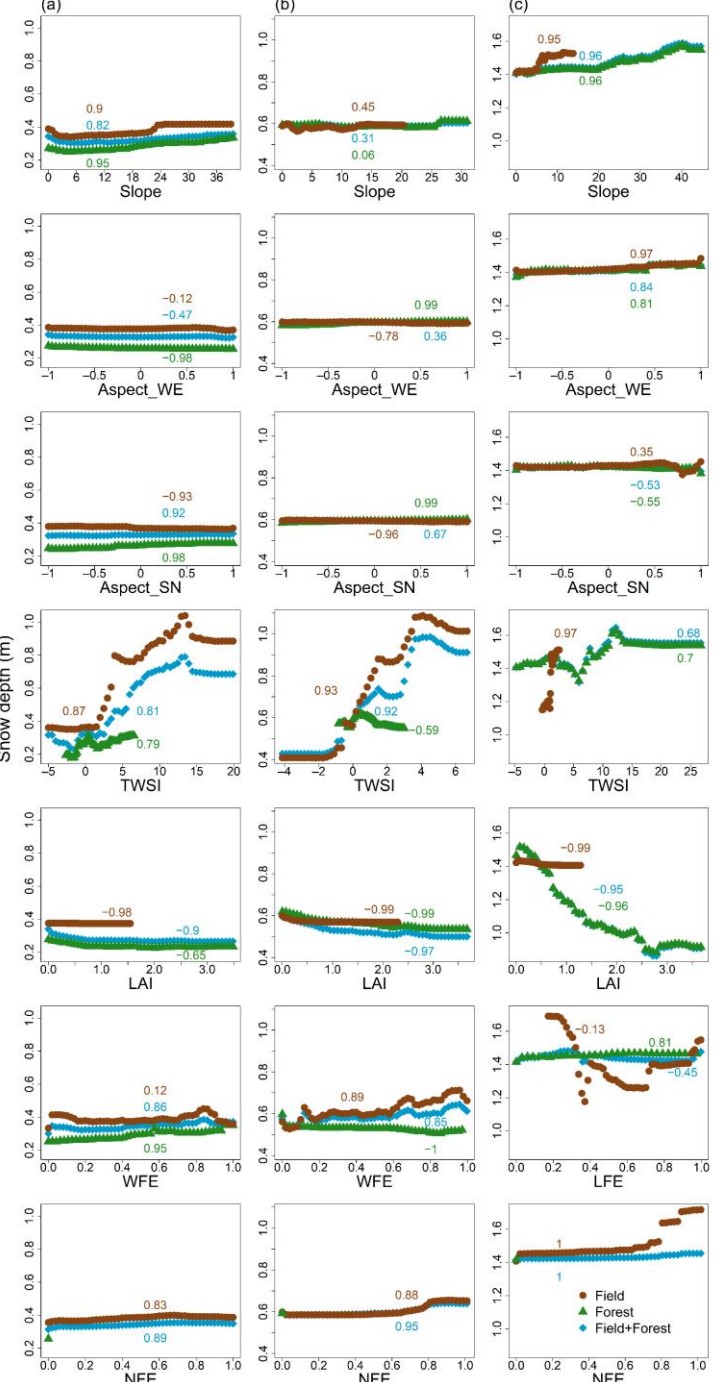

**Figure 8.** Partial relationship of landscape predictor variables with snow depth. **(a)** Sainte-Marthe, **(b)** Saint-Maurice and **(c)** Montmorency. Predictor variables are presented by rows and sites by columns.

### 3.3.4    Performance of RF models at each site

Figure 9 displays the RF model estimates versus observed snow depth with corresponding OOB statistics for each site. Statistics are presented individually for the field, forest, and full domain (field+forest). Among the three sites, Sainte-Marthe RF model generally performs better with an OOB $R^2$ of 0.66 and RMSE of 0.083 m, and Montmorency shows the weakest performance with an $R^2$ of 0.30 and RMSE of 0.261 m. All field models perform comparatively better with higher $R^2$ and lower RMSEs values than their forest models.

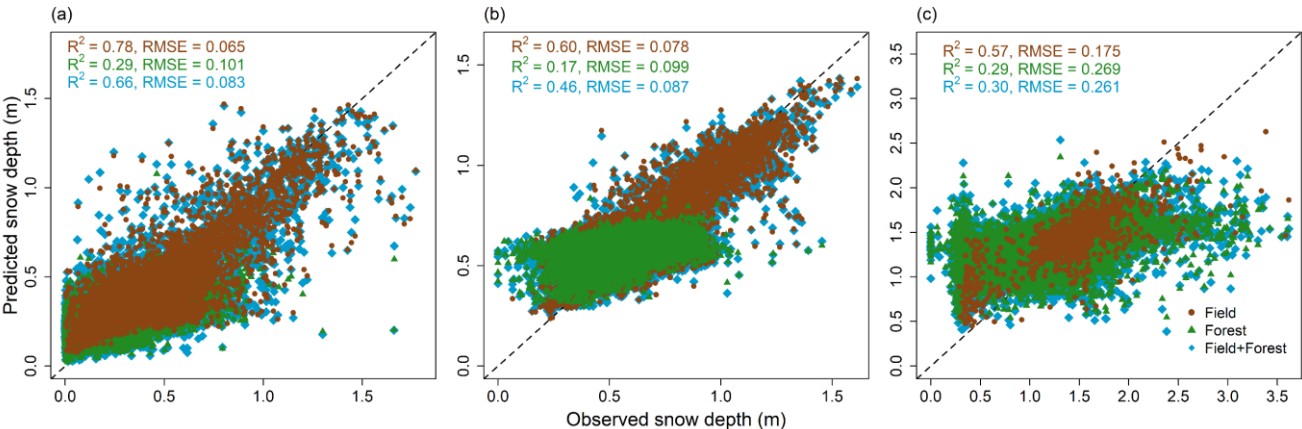

**Figure 9.** RF model performance against observed snow depths. **(a)** Sainte-Marthe, **(b)** Saint-Maurice and **(c)** Montmorency. The stippled line depicts the 1:1 relationship.

## 4    Discussion

### 4.1  Spatial variability of forest versus field snow depths

Snow depths in Fig. 4 show remarkable microtopographic variability across all sites. Our results in Sainte-Marthe underpin the previous finding that forested areas accumulate less snow than the adjacent open areas due to canopy interception and sublimation losses and sheltering from wind (Pomeroy and Granger, 1997; Hopkinson et al., 2004; Varhola et al., 2010a; Zheng et al., 2018; Hojatimalekshah et al., 2021). But the other two sites show on average, a similar amount of snow accumulation in the field and forest. The dense coniferous canopy cover in Montmorency prevented laser shots from reaching the ground at some locations and consequently resulted in data gaps in the snow depth map (Fig. 4c). The snow depth patterns in the coniferous site thus appear to be dominated by canopy closure, i.e., forest clearings have higher snow depths than adjacent canopies. Such patterns have been previously reported by both ALS and UAV-lidar studies in western alpine/pre-alpine environments with different climates (Hopkinson et al., 2004; Zheng et al., 2016; Mazzotti et al., 2019; Jacobs et al., 2021). The overall amount of snow in the forest compared to field in the boreal forest of Montmorency could thus be underestimated due to poor lidar coverage under dense canopies.

At the agro-forested sites, the comparatively higher snow depths observed in the open field compared to the adjacent forest patches are in contrast to what Aygün et al. (2020) observed in similar environments in southern Québec. They measured a lower snow accumulation in exposed agricultural fields (excluding the canals and the forest edge) compared to the adjacent deciduous and mixed forests. Our results show that the higher snow depths at the two agro-forested sites principally correspond to canals and streams in the field and the forest edge, which trap the snow blown from the open field with greater fetches. Hence canals/streams and forest edges constitute the main structuring elements of snow spatial variability at these sites. However, if canals and forest edge snow depths are discarded, the agro-forested snow depth maps illustrate a somewhat similar phenomena to Aygün et al. (2020), where snow depths in the exposed field are slightly lower than those in the forest. In Saint-Maurice, clusters of high snow depth values in the central area of the field in Fig. 4b could be due to local redeposition of snow by the wind in the microtopography, or larger-scale effects. This could not be verified as unfortunately, the manual measurements in Saint-Maurice could not be retrieved due to a probe malfunctioning (Dharmadasa et al., 2022). Yet, the *TWSI* map (Supplement Fig. S3) suggests that microtopographic wind sheltering could be the reason for the local snow deposition closer to the forest edge. The probable cause for the other larger high snow depth cluster between the two streams in the field could not be explained from the available predictors. They could be explained by the influence of the narrow riparian strips of bushes and shrubs surrounding the canals on blowing snow redistribution. Ultimately, as canopy interception and losses in deciduous and mixed forests are expected to be small (Hopkinson et al., 2010; Aygün et al., 2020), the amount of differential snow depths between the open field and forest would mostly depend on the amount of erosion in the field, and perhaps snowmelt losses in the open field prior to peak snow accumulation. Moreover, the snow depth maps suggests that the redistribution of eroded snow in fields along the forest edges is a prime process in agro-forested landscapes.

## 4.2 Scaling characteristics of forest versus field snow depths

### 4.2.1 Omnidirectional semi-variograms analysis

Omnidirectional semi-variogram analyses revealed distinct scaling behaviors in forest versus field snow depths (Fig. 5). Our results suggest a more variable (high semi-variance values) and more spatially continuous (larger scale break distance) snowpack in the Montmorency boreal forest compared to the temperate forest sites. The snowpack in the mixed forest at Saint-Maurice was less variable and more spatially continuous than that in the Sainte-Marthe deciduous forest. Compared to forested areas, the snowpack in field areas was less variable and more spatially continuous. Shorter scale break distances in forested areas compared to open field areas (Fig. 5) is analogous to previous studies that studied the fractal distribution of snow depths with lidar data. These studies reported scale break distances of 4 m for a shrub-dominated sparsely distributed subalpine site (Mendoza et al., 2020b), 7–9 m for high to moderately dense coniferous forests (Trujillo et al., 2007; Trujillo et al., 2009), 12 m for a moderately dense deciduous forest (Trujillo et al., 2007; Trujillo et al., 2009), 15.5 m for a dense coniferous forest with open meadows (Deems et al., 2006; Fassnacht and Deems, 2006), and 16.5 m for a sparse coniferous forest (Deems et al., 2006; Fassnacht and Deems, 2006). We found the shortest scale break distance of 4.4 m for the dense deciduous forest in

Sainte-Marthe, an intermediate distance of 5 m for the moderately dense mixed forest in Saint-Maurice, and a value of 6.5 m

for the dense coniferous forest interspersed with gaps in Montmorency. These values are rather smaller than those reported by previous studies, except Mendoza et al. (2020b). This could be due to structural characteristics of the forests such as canopy density and size of open areas (gaps). It is also plausible that the dense point cloud provided by UAV (~150–600 points m$^{-2}$: Zhang et al., 2019; Harder et al., 2020; Jacobs et al., 2021; Dharmadasa et al., 2022) was able to resolve spatially distributed snow depth patterns at finer scales than that permitted by previous ALS surveys, which had typical point densities of ~8–16

points m$^{-2}$ (Kirchner et al., 2014; Broxton et al., 2015; Broxton et al., 2019; Currier et al., 2019). However, similar to the findings reported by Deems et al. (2006) and Trujillo et al. (2007) our topography+vegetation surface data show scale break distances at the same order of magnitude as the forest snow depths at all sites. This indicates that the variability of vegetation (trees) governs the pattern of snow deposition and distribution within the forest (Deems et al., 2006).

The relatively higher scale break distance in Montmorency forest snow depth could be due to the prevailing large gaps in the

forest as a result of silvicultural practices and the higher efficient canopy interception of conifers. Coniferous trees have a substantial impact on snow depths as they intercept snow efficiently and unload it around the crown (Zheng et al., 2019). Thus, a longer correlation length (at least the diameter of a tree crown) is expected as well as greater variability of snow depth in coniferous environments compared to the more random deciduous tree structures which have reduced and more transient snow storage (Mendoza et al., 2020b). Leafless deciduous trees aid faster unloading of snow through branches as opposed to

unloading around the crown in conifers and thus would result in a smaller correlation length in snow depth.

The difference in scale break distances in field snow depths compared to bare earth topography indicates that the bare ground surface in field areas was certainly altered by the snow accumulation. In Sainte-Marthe, snow accumulation increases the roughness of the bare ground whereas, in Saint-Maurice, snow accumulation results in a smooth surface compared to the ground underneath. i.e., interactions of snow with bare ground in Sainte-Marthe field change the scale invariance behavior to

multi-scaling, and in Saint-Maurice, these interactions smooth the surface and resulted in a larger scale break distance than that of the bare ground. However, the larger scale break distance and more gentle slope of the Sainte-Marthe field semi-variogram (Fig. 5a) compared to Saint-Maurice (Fig. 5b) suggests that the snowpack in Sainte-Marthe field is still smoother and more spatially continuous than that of Saint-Maurice. This interpretation is supported by the snow depth map in Fig. 4a, which shows a smooth snow depth pattern that is only disrupted by preferential accumulation within irrigation canals/streams.

In Montmorency field, rather than interactions of snow with bare ground, the meteorological station network appears to modify the snow accumulation and distribution patterns and resulted in different multi-scaling behavior than the bare ground. In general, large scale break distances (11–14 m) compared to forested areas were found in field snow depths at all sites except the short, first scale break distance (5.8 m) in Montmorency. With the absence of vegetation in the field in winter and its high exposure to wind at the two agro-forested sites (Fig. 2a, b), these values are of similar magnitude to those reported for wind-

exposed slopes in alpine environments (13.8–20.5 m) by Schirmer and Lehning (2011), Mott et al. (2011), Mendoza et al. (2020a), and Mendoza et al. (2020b). In the Montmorency field, mostly sheltered from the wind, the short and large scale

break distances could be due to the influence of preferential snow accumulation near the meteorological equipment (e.g., concentric snow accumulations patterns around the two double-fenced precipitation gauges in Fig. 4c).

Generally, the scale break distances found in this study suggest that the scale selected for modeling or sampling in similar environments should be well below these values, in order to represent the small-scale variability of the snow depth.

### 4.2.2 Directional semi-variograms analysis

Sainte-Marthe field snow depths did not show any directionality, most probably as a result of the interactions of snow with two dominant and opposed wind directions. In contrast, Saint-Maurice field snow depths showed anisotropic behaviors along and perpendicular to the dominant wind direction. Narrow anisotropic patterns perpendicular to the dominant wind direction are due to the snow accumulation alongside canals. Even though the canals were discarded in semi-variogram analysis, as seen from Fig. 4b, snow accumulation alongside the canal banks up to a few meters into the field is still significant. Broader anisotropic patterns along the dominant wind direction are due to the influence of wind. This directionality is also shown in the snow depth map in Fig. 4b, where the change of snow depth values along the direction perpendicular (northwest-southeast) to the dominant wind direction is more drastic than the change of snow depths along the dominant wind direction towards the forest. However, forest snow depths at both agro-forested sites show anisotropic behavior, although not very strong, parallel to dominant wind directions. This indicates an influence of blowing snow on the snow distribution patterns in the forest, and hence a possible penetration of blowing snow from field to forest. The isotropic behavior in the Montmorency field and forest, on the other way, is not surprising given that the site is sheltered from the dominant winds (Fig. 2c).

### 4.3 Relationship of snow depth to topographic and vegetation characteristics

During the analysis, some consistent patterns emerged between all three sites, which have been found in previous studies, i.e., snow depth generally decreases with an increase in *LAI* values (Varhola et al., 2010b), and snow depth increases with an increase of *TWSI* (Revuelto et al., 2014).

### 4.3.1 At the agro-forested sites

At the two agro-forested sites, field snow depth variability is governed by preferential snow accumulation in canals/streams and the microtopography of the local terrain. As such, the highest wind sheltering values were found in canals/streams which accumulated more snow (Fig. 4 and Supplement Fig. S2, S3). Within the forested areas, the influence of forest structure (*LAI*) was not as strong as expected; instead, the influence of microtopography appeared to be mostly governing the snow depth variability. The lower influence of *LAI* at these sites probably reflects the abundance of leafless trees in winter, which reduce interception losses and concurrent spatial snowpack variability. Moreover, the microtopography of these landscapes is closely related to the surficial geology of the sites. Preserved forested patches in the St. Lawrence River lowlands often correspond to less favorable soil conditions, such as glacial till and/or bedrock outcrops and associated rougher microtopography. Conversely, agricultural fields are developed on glaciomarine or fluvioglacial sediments that are flatter in nature and also

leveled by machinery (MFFP, Québec Research and Development Institute for the Agri-Environment (IRDA) and La Financière Agricole du Québec (FADQ)). Under limited wind transport, the rougher microtopography in forests creates a

directional bias that promotes lateral transport of snow particles (bounce/ roll/ ejection) and therefore enhances the smoothing of the snow surface (Filhol and Sturm, 2019) which dominates the snow heterogeneity within the forest. The absence of apparent preferential snow accumulation on different slope orientations in agricultural fields suggests a smoothening of the topography by the snow cover due to wind redistribution in the field. The more rugged microtopography of the forested soil on the other hand seems to be preserved and to influence the snow cover through differential radiation loading, resulting in

more snow accumulations on northerly slopes in the forest compared to that in the field (Fig. 8a, b).

At the landscape scale (field+forest), the agro-forested sites are dominated by blowing snow accumulation along the forest edges (Fig. 7a, b). This effect is well visible on the lidar-derived snow depth maps too (Fig. 4a, b). Comparatively high wind speeds and more constrained dominant wind directions (Fig. 2a, b) at these sites create favorable conditions for preferential deposition of blowing snow at the forest edge due to the large expanses of open terrain upwind of the windward forest edges.

Preferential snow deposition by wind-induced snow drifting along the forest edge has been previously reported in alpine environments by Veatch et al. (2009), Essery et al. (2009), Broxton et al. (2015), and Currier and Lundquist (2018). However, there seems to be only limited penetration of blowing snow inside the forest in windward directions (*WFE* forest points in Fig. 8a, b and Fig. 6d, e).

Shading by the forest edge seemingly does not have a significant influence on the snow depth variability at these sites during

the accumulation season. Shading effects would however probably have some influence on snow depth patterns during the melting season (Hojatimalekshah et al., 2021). The spatial heterogeneity of snow depths and associated processes challenge distributed snow modeling using hydrologic response units (HRUs) in agro-forested landscapes (Aygün et al., 2020), where HRUs are classified as field and forest patches but disregard boundary effects. Aygün et al., (2020) successfully modelled (Nash–Sutcliffe efficiency of 0.57 over 23-year simulation of SWE) blowing snow transport in fields and the preferential

accumulation in canals and streams, and assumed that once these were filled, any further blown snow accumulated in the forest. Our results confirm the preferential accumulation in field canals and streams but suggest that further blown snow first preferentially accumulates at the forest edge, which should eventually be represented as distinct HRUs in distributed hydrological models of agro-forested landscapes.

### 4.3.2    At the boreal forested site

The findings in agro-forested sites are in contrast with the boreal forested environment, where forest structure (*LAI*) predominates on the variability of snow depth (Fig. 7 and Fig. 8). The small field appears to have fewer microtopographic features and is mostly sheltered from the most frequent winds coming from the northwest direction (Fig. 2c). The relatively greater positive *TWSI* values at this site compared to agro-forested sites imply more rugged microtopography and a larger degree of wind sheltering in the forested terrain (Fig. 8c and Supplement Fig. S4). However, since wind is mostly impeded by

the coniferous trees, the *TWSI*-snow depth relationship in the forest suggests that the snow displacement is driven by small-

scale bounce/ejection/roll mechanisms, and preferential snow deposition is driven by immobilizing mechanisms such as adhesion, cohesion, and physical interlocking of snow particles (Filhol and Sturm, 2019) and unloading of snow by the canopy (Zheng et al., 2019). The lesser importance of *TWSI* (0.41 compared to 0.97 of *LAI*, the dominant predictor, Fig. 7) as snow depth predictor in the coniferous forest compared to deciduous (*TWSI* = 0.74, the dominant predictor) and mixed (*TWSI* of 0.25 compared to 0.33 of *WFE*, the dominant predictor) forests, and the more or less constant snow depth values at higher *TWSI* values (Fig. 8c) suggest that microtopography has a more restricted influence on deeper snowpack at this site compared to the shallower snowpack at the agro-forested sites. In other words, in the absence of wind, increasing snow depths reduce/inhibit surface undulations and promote more spatially continuous snow cover (Filhol and Sturm, 2019). The spatial arrangement of the trees may have a larger control on snow depths in the boreal forest, i.e., forest gaps in the coniferous forest with various slopes and aspects create pronounced and distinct snow depth variabilities inside the forest (Woods et al., 2006). For instance, in Montmorency, superimposed *TWSI* and *LAI* maps (Supplement Fig. S4) show that the high snow depth values associated with *TWSI* values of 10–12 (Fig. 8c) are associated with a forest gap that likely prevents snow interception and accumulates more snow. Our results support the findings of previous studies that the snow depth distribution in coniferous environments is mainly governed by the canopy characteristics such as structure, distribution, and type of vegetation (Winkler et al., 2005; López-Moreno and Latron, 2008; Varhola et al., 2010a; Zheng et al., 2018; Safa et al., 2021; Koutantou et al., 2022). Our findings however show that the microtopography, even under wind-sheltered conditions in the forest, still plays an important part of the spatial variability.

### 4.3.3 At the single-tree scale

At the single-tree scale, the variability of terrain and vegetation characteristics within single tree canopies is supressed. Our results show that masking this intra-tree variability reduces the predictive power of RF models (lower $R^2$, Supplement Fig. S5), and also changes the variable importance (Supplement Fig. S6) to some extent. However, even at the single-tree scale, wind-related forest edge effects and microtopography still explain a major portion of snow spatial variability at agro-forested sites, while at the boreal forested site, canopy characteristics and microtopography have the highest impact (Supplement Fig. S6). The most noteworthy difference in landscape influence on snow depth variability comes from the slope and aspect, which show a more pronounced impact on snow depth at this scale (Supplement Fig. S7). For instance, all the forested areas show clear signs of snow accumulation on northerly oriented slopes (*Aspect_SN* in Supplement Fig. S7). The influence of the various meteorological stations in the Montmorency field could be the reason for prominent preferential snow accumulation on the southern slopes at this site. However, this analysis shows that at scales larger than the scale break distances at these sites, large-scale topographic characteristics like slope and aspect play a more significant role in shaping snow accumulation and distribution patterns than that at smaller, intra-canopy scales.

### 4.4 Comparison of RF model performances

#### 4.4.1 Comparison between the sites

Our RF model showed variable performances, with overall OOB $R^2$ of 0.30–0.66 (Fig. 9). All sites have different climates. The higher performance at Sainte-Marthe could be due to a combination of different factors. Early snowmelt due to frequent rain-on-snow events in this region (Paquotte and Baraer, 2021) and the presence of basal ice as observed in the field campaigns might have contributed to a more structured snowpack in the Sainte-Marthe forest and hence improved the prediction of snow depth compared to the other agro-forested Saint-Maurice site. The high $R^2$ values in fields at all sites (0.78 in Sainte-Marthe, 0.60 in Saint-Maurice, and 0.57 in Montmorency) indicate that the models captured the most relevant processes through the predictor variables considered. In contrast, Saint-Maurice forest had the worst performance (0.17). This could be due to underlying processes/variables not considered in our model, possibly associated with the canopy structure of the mixed forest. Moreover, the reduced sampling under coniferous trees due to limited lidar penetration could also have affected grid-scale mean snow depth and resulting relationships with landscape metrics in the Montmorency forest.

#### 4.4.2 Comparison with previous studies

The previous studies that used RF models to estimate snow depths/SWE (Bair et al., 2018; Yang et al., 2020) were mainly focused on mountainous watersheds with large elevation gradients and with less or no vegetation and reported average Nash–Sutcliffe efficiencies as high as ~0.7 and RMSEs of 44–73 mm, where the major part of this variance was explained by elevation. Safa et al. (2021) developed site-specific RF models to predict snow-covered areas using vegetation density, average incoming shortwave, and longwave radiation, total precipitation, and average air temperature and reported mean absolute errors of 0.05–0.12 m in mixed coniferous sites. In addition, the abundance of studies that employed MLR (Jost et al., 2007; Lehning et al., 2011; Grünewald et al., 2013; Revuelto et al., 2014; Fujihara et al., 2017) and BRT (Winstral et al., 2002; Anderton et al., 2004; Molotch et al., 2005; Revuelto et al., 2014) in alpine environments with rocky outcrops and pasture or no vegetation also reported $R^2$ of 0.25–0.91 where a substantial portion of the snow depth variability was explained by terrain parameters, mostly elevation. However, model performances are shown to be degraded with the presence of forests. Studies conducted in forested terrain with relatively small elevation ranges reported $R^2$ of 0.25–0.51 by MLR (Zheng et al., 2016; Zheng et al., 2018) and BRT (Erxleben et al., 2002; Veatch et al., 2009; Baños et al., 2011). Musselman et al. (2008) proved that including detailed vegetation information like micro-scale vegetation-induced solar radiation, distance to the canopy, and tree bole could improve BRT performance to 0.68 in a forested area. Compared to previous works in forested terrain, we believe our model fits (overall $R^2$ of 0.30–0.66) are in a reasonable range.

#### 4.4.3 Comparison to MLR models

The relatively good success of MLR in previous studies to study landscape control on snow accumulation is mostly attributed to elevational controls on snow accumulation, i.e., orographic enhancement of precipitation gradient and adiabatic cooling

which promotes higher snowfall fraction and reduced ablation at higher elevations. However, in low elevation landscapes, more complex relationships are expected between snow depths, vegetation, and topography, which would likely be poorly captured by linear relationships. As shown in Table 3, our RF models show a significant improvement with higher $R^2$ and

lower RMSE values compared to MLR models at all sites. Since the MLR models at each site were developed using the same predictors described in section 3.3.1., this suggests the deficiency of MLR models in capturing the underlying processes at these sites. Figure 8 shows that almost all variables have a nonlinear relationship with snow depth, which linear models are unable to capture. Our RF results thus highlight the importance of considering this nonlinearity in statistical models, as RF notably allows capturing nonlinear relationships between snow accumulation and landscape variables, while protecting against

the typical overfitting of single decision trees.

**Table 3.** Comparison of RF and MLR model performances of study sites

| RF | | | | | | |
|---|---|---|---|---|---|---|
| | $R^2$ | | | RMSE | | |
| | Field | Forest | Field+Forest | Field | Forest | Field+Forest |
| Sainte-Marthe | 0.78 | 0.29 | 0.66 | 0.07 | 0.10 | 0.08 |
| Saint-Maurice | 0.60 | 0.17 | 0.46 | 0.08 | 0.10 | 0.09 |
| Montmorency | 0.57 | 0.29 | 0.30 | 0.18 | 0.27 | 0.26 |
| MLR | | | | | | |
| | $R^2$ | | | RMSE | | |
| | Field | Forest | Field+Forest | Field | Forest | Field+Forest |
| Sainte-Marthe | 0.18 | 0.04 | 0.21 | 0.12 | 0.12 | 0.13 |
| Saint-Maurice | 0.32 | 0.08 | 0.17 | 0.10 | 0.10 | 0.11 |
| Montmorency | 0.02 | 0.13 | 0.12 | 0.26 | 0.30 | 0.29 |

## 4.5 Note on potential variables/predictors in similar landscapes

One particularity of our sites (also related to the scale of the analysis) is the negligible elevation range. Many studies conducted in mountainous environments have shown the preponderant influence of elevation on the distribution of snow cover. While

the elevation range becomes important over a larger extent on the Canadian shield (Montmorency-type physiography), the low elevation St. Lawrence lowlands (Sainte-Marthe and Saint-Maurice) remain mostly flat, and local topography (terrain roughness) and land cover and land use are expected to control the spatial distribution of the snow cover. As confirmed by our results, in agro-forested land covers, wind-related forest edge effects will also have a substantial impact on snow deposition, and distribution patterns.

## 4.6 Limitations of the study

This study provides insight into the scaling properties of the snowpack and the effect of different topographic, vegetation, and forest edge characteristics on snow depth variability in open versus forested areas with different canopy covers. However,

there are potential limitations with some of the methods presented in this study. For instance, despite our efforts to incorporate processes/variables influencing the spatial distribution of snow depths with available data, the comparatively lower

performance of RF models in Saint-Maurice and Montmorency indicates that there could still be some processes/variables that were unable to accounted for (e.g., soil parameters, snowpack state, and meteorological variables). Another limitation comes from the unexplained snow depth variability that is within the UAV-lidar system detection limit. Especially in Montmorency, there were observation gaps by UAV-lidar due to the thick canopy cover that eventually affected the accuracy of snow depth and ground surfaces rasters and derived landscape descriptors (e.g., slope, LAI, etc.). The dominant predictors identified in

this study might also depend on the timing of the survey date (e.g., near peak snow accumulation versus early and mid-winter, or during the melt period). Hence, repeat surveys with UAV-lidar to track the temporal evolution of the snowpack would be required to fully address this question in the future. However, the analysis presented here is thought to largely reflect the typical conditions at the sites and to portray key differences between agro-forested and boreal landscapes.

## 5   Conclusions

In this study, including wind-related forest edge effects in agro-forested sites and incorporating canopy characteristics in the coniferous site increased the statistical prediction accuracy of snow depth spatial variability by more than 90 %. This implies the importance of including and better representing these processes in process-based models. Taken together, our results suggest that in agro-forested landscapes of the St. Lawrence valley, geomorphological assemblages drive the differential snow accumulation between field and forested areas, i.e., rugged glacial deposits with preserved forests favor more snow

accumulation whereas flat glaciomarine sediments in the exposed fields promote snow erosion. The blowing snow redistributed from the fields gets trapped in canals/streams and accumulates along the forest edges, accounting for the highest local snow depths in these landscapes. Furthermore, within deciduous/mixed forests, it is rather the underlying topography and/or the forest edges that govern the snow depth variability, while within the coniferous environment, it is the forest structure variability. More often, these processes are not fully represented in process-based models. For instance, most of the process-

based models like CRHM (Pomeroy et al., 2007), and SnowModel (Liston and Sturm, 1998) prescribe a single, typical LAI for land cover classes. This ignores the variability within stands which could compromise larger scale estimates of snowpacks. The recent development of hyper-resolution process-based models does account for fine scale canopy structure (Mazzotti et al., 2020a; Mazzotti et al., 2020b), yet representing microtopographic characteristics like terrain roughness is still problematic. Our results suggest that snow redistribution at forest edges, spatial variability of forest structure, and better representation of

microtopography and prominent topographical features such as canals are important processes/variables that should be taken into account in process-based models. This highlights the advantage of using high resolution data to characterize small-scale processes and therefore explicitly resolve snow depth variability.

In addition, since the selected sites are representative of typical agro-forested and boreal landscapes in southern Québec, the findings of this study could be applied/extrapolated to similar landscapes in the region and any similar environments where

similar processes operate. It is worth noting that future efforts in designing modeling parameterizations that include forest edge effects would benefit from incorporating the meteorological conditions together with topographic and vegetation characteristics.

**Author Contributions:** Conceptualization, C.K.; methodology, C.K. and V.D.; formal analysis, V.D. and C.K.; data curation, V.D.; writing-original draft preparation, V.D.; writing-review and editing, C.K. and M.B.; supervision, C.K. and M.B.; project administration, C.K.; funding acquisition, C.K.

**Funding:** This study was financially supported by the Canada Research Chair program (grant number 231380) and the Natural Sciences and Engineering Research Council of Canada (NSERC discovery grant CRSNG-RGPIN-2015-03844) (Christophe Kinnard) and a doctoral scholarship from the Centre de Recherche sur les Interactions Bassins Versants-Écosytèmes Aquatiques (RIVE, Vasana Dharmadasa).

**Data Availability Statement:** The data presented in this study are available on a reasonable request from the corresponding author.

**Acknowledgments:** The authors extend their appreciation to the members of GlacioLab for their help during our fieldwork. Moreover, the authors are grateful to the Sainte-Marthe municipality, Québec, Canada and members of NEIGE_FM, Forêt Montmorency, Québec, Canada.

**Conflicts of Interest:** The authors declare no conflict of interest.

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
