# Peer review of "Figure S1. Sainte-Marthe snow depth and predictor variables maps. Elevation is presented as ellipsoidal heights."

_The Cryosphere, 2022_

## Referee Comment (RC3)

Major comments:
1. This manuscript is using UAV-lidar data to understand the snow-depth heterogeneity in agro-forested environments and boreal forests. Since the author also used the slope and aspect as topographical variables for studying the snow-depth distribution. However, at site Saint-Marthe and Saint-Maurice, the elevation difference is actually very small comparing to the spatial scale of the study area, it would be kind of surprising to observe any meaning effects from topographical variables in these 2 study areas. And although Montmorency has a elevation difference of 20 meters in the study area, it seems the main area is facing towards south east direction. It would also be helpful to visualize the distribution of the slope and aspect of the 3 study areas to make sure there is significant variability in these predictors of snow-depth before feeding them into a model like random forest.
2. It is not very clear on how the forest edge descriptors are derived, it would be helpful to have a visualized illustration to demonstrate how variables are derived based on canopy cover data from lidar.
3. The 2$^{nd}$ objective raised in the Introduction part for this manuscript seems to be extremely open ended. Is there a particular hypothesis the authors would like to test with the used dataset and validate the hypothesis throughout the manuscript? The current objective of "exploring the relationship between snow depth, topography, and forest structure" seems too vague and not specific enough.

Minor comments;
Figure 1 – it looks like the first 2 study areas are very flat with low elevation and the 3$^{rd}$ one has elevation difference and the elevation is much higher. It the precipitation in this area affected by orographic effect as well?

Line 154-155: given it is 1 m diagonal cross shape, why it is 1.4x1.4 m grid cell? Isn't it going to be 0.7m x 0.7m grid cell instead?

Line 162-165: it is not clear why UAV-lidar is more robust and the technology represents an improvement to previous studies.

Line 184-186: The closest weather station is 19 km away from Saint-Maurice. Is the wind data going to be trustable for this site given it is very far and the wind speed and direction can be quite different comparing to the actual on the site, right?

Line 193: how is LAI, CC, and GF calculated? By using Lidar360 software?

Line 202: why the grid size for vegetation is so much larger than the resolution of the snow-depth (1.4x1.4 m). It seems the vegetation grid resoluation is so much coarser and are we able to capture all the forest variable based on such a low resolution?

Section 2.2.4, please see major comment #2, it is not very clear how d and d_max are derived based on the forest-covered lidar data.

Line 250-253, are hyperparameters in Random Forest tuned or selected before training each model?

Line 257: It is not very clear how forested vs. fields are defined. It would be helpful to have a map of these site showing this binary variable. And why don't we use this binary variable directly in the RF model directly? Is it to show at different area how other variables affecting snow-depth differently?

Figure 3: it is a bit surprised to me at Montmorency there is not many data points for under canopy. Then we might not be able to observe a lot of under canopy snow-depth signals.

Figure 4: how is the scale break selected? Please describe that in the Method section.

Figure 5: it might be better to use bar chart with different colors. It is a bit difficult to differentiate color the marker styles on this scatter plot.

Section 3.3.3: there is only one line in the Method section (line 267) discussed about the partial relationships of preditor variables with snow depth. It is still not very how that is calculated. Please add details in the Method section.

Figure 7, it looks like the model is not performing very well at Saint-Maurice and Montmorency. The slope of the predictioned vs. observed is not close to 1. What would be the reason that the trained RF model is underfitting and has this systematic bias?

---

## Author Comment (AC1)

We thank the anonymous referee #1 for his/her thorough review with constructive comments and suggestions that certainly will improve the manuscript. In the following, we will address the referees' comments point by point. We mark "black" the comments given by the referee, and our responses in "blue".

**Comment on tc-2022-124**
Anonymous Referee #1
Referee comment on "Topographic and vegetation controls of the spatial distribution of snow depth in agro-forested environments by UAV-lidar" by Vasana Dharmadasa et al., The Cryosphere Discuss., https://doi.org/10.5194/tc-2022-124-RC1, 2022

**Summary**

In this paper, the authors analyze the scaling properties of lidar-derived snow depth and possible dependencies with topographic and vegetation descriptors in two agro-forested and one coniferous site in eastern Canada. They conduct variogram analyses on snow depth fields to find possible scale break lengths that define regions with self-similar behavior, and develop random forest models to characterize predictor importance. The results show scale breaks spanning 4-7 m in forested sites, and relatively longer values in field areas – up to 18 m in wind exposed fields – in agreement with previous studies. The results also show that wind-related forest edge descriptors mostly explain snow depth variability in agro-forested sites, while canopy characteristics (i.e., forest structure) are more important in the coniferous site.

Overall, the topic, research questions and experimental setup are interesting for the snow hydrology community. The literature review and discussions are quite extensive (maybe more than needed), and the graphics included in the manuscript are very nice. I have three major comments that I think the authors should address before this paper is considered for publication. Additionally, the authors will find a set of minor comments and editorial suggestions that may be helpful to improve the quality of this manuscript.

We thank referee #1 for the remarks and overall positive assessment.

**Major comments**

1.  Fractal analysis:

i.  It is not clear from Figure 4 that scale breaks actually exist and, therefore, all the remaining analyses and interpretations remain unsupported, unless the authors demonstrate quantitatively that the snow depth scaling behavior changes. I recommend the authors to revise Mendoza et al. (2020a) as a reference on how to detect scale breaks in variogram analysis.

We acknowledge this matter that also resonates in other reviewer's comments, that we need to quantitively demonstrate how the scale breaks were identified. Therefore, we repeated the variogram analyses following the method described in Mendoza et al. (2020), as suggested.

ii. The authors need to show quantitatively that variograms hold a power law (which is required to indicate that a spatial pattern is actually fractal). I think that, at the very least, the authors should demonstrate that a linear model in the log-log space holds before and after the scale break, with a high coefficient of determination (e.g., $R^2 \ge 0.9$). I also recommend the authors to test whether other geostatistical models are more suitable for this data (e.g., spherical, Gaussian).

We repeated the variogram analysis of snow depths at all sites by considering the reviewer's recommendation. The following steps were performed to identify scale breaks:

- A change point analysis was conducted on variograms in log-log space using the ecp package in R to identify possible break points. i.e., possible break points indicate that each cluster of variogram points separated by these break points share a similar trend.
- Then, a piecewise linear regression was fitted to the variogram points by using the candidate break points identified in the previous step. This function plots the linear models in variograms only when there is a significant trend ($p < 0.05$).
- After confirming the existence of clusters of points with a significant trend, linear least square regression models were fit in log-log space for each cluster of variogram points identified in step 2.
- Checked whether the changes in the slopes of the log-log linear models are larger than 20% and that the 95% confidence limits of the slopes do not overlap. Inspected visually in variogram of this scale break. Verify the $R^2$ are greater than 90%. If all these conditions are fulfilled, the existence of a scale break is confirmed.

We replotted the variograms with the updated scale break distances (some of the scale break distances were slightly changed) and linear regressions. Figure 4 will be replaced with the updated plot (which is placed under response iii) in the revised manuscript.

iii.   The authors might consider comparing omnidirectional variograms of snow depth (and potential scale breaks) with those obtained from bare earth topography and topography+trees (e.g., Deems et al. 2006; Trujillo et al. 2007). Additionally, the analyses could be enriched by computing directional variograms and associated scaling parameters, in order to establish possible connections with dominant wind directions (e.g., Deems et al. 2006; Schirmer and Lehning 2011; Clemenzi et al. 2018; Mendoza et al. 2020a).

Considering the referee's comment, we compared the omnidirectional variograms of the snow depths with those obtained from bare earth topography and topography+vegetation. The following plot shows the updated variograms that will replace Figure 4 in the current manuscript. Generally, it is noticeable that the scale break distances in the forested areas are in the same order of that in topography+vegetation variograms. Scale break distances in bare earth topography are altered by snow accumulation. These observations will be discussed in the revised manuscript accordingly.

[Figure]

Omnidirectional semi-variogram for the field and forested areas for (a) Sainte-Marthe snow depth, (b) Saint-Maurice snow depth, (c) Montmorency snow depth, (d) Sainte-Marthe bare earth topography and topography+vegetation, (e) Saint-Maurice bare earth topography and topography+vegetation, and (f) Montmorency bare earth topography and topography+vegetation. In the figure, Topo denotes bare earth topography and Topo+veg denotes topography+vegetation.Vertical lines indicate the dominant scale breaks.

Additionally, we will compute the directional variograms and discuss the isotropic vs anisotropic behavior of snow depth in field and forested areas in the revised manuscript. A new plot with directional variograms will be added to the revised manuscript.

We think these amendments and additions will significantly improve the quality of the spatial correlation analysis section of the paper.

2.   Partial relationships: I think the authors should be more quantitative, since the reasoning provided in section 3.3.3 is subjective and difficult to follow. You can easily improve this section by computing the Spearman rank correlation coefficient, and reporting the p-values. I suggest avoiding statements like 'strong relationships', 'stronger that', 'slight decrease with increasing', etc.; instead, you can show the numbers and let the readers judge.

Thank you for this suggestion. We will calculate the Spearman rank correlation coefficient and update all the plots and change the discussion accordingly in the revised paper.

3.   The authors may consider using the following sequence to display RF results:

i.   Exploratory analyses with scatter plots of snow depth vs. predictors (current Figure 6). In any case, I recommend the authors verifying these results, since it's very odd that there is practically no scatter along the y axis. Are you displaying all points within your domains in each panel?

Figure 6 displayed partial dependence plots of snow depth against each independent variable, and not simple bivariate scatter plots (which would scatter along the y axis). Partial dependence functions are typically used to help interpret models produced by machine learning models such as random forest analysis (Jerome, 2001). It is a risk adjusted alternative to variable dependence. We used the pdp package in R to obtain the partial dependence of the variables in random forest models (Greenwell, 2017). Here, each partial plot was generated by integrating out the effects of all variables beside the covariate of interest. Partial dependence data in each plot were constructed by selecting points evenly spaced along the distribution of the variable of interest. This subsampling helps to cut down computational time substantially. We used the default subsampling of 51 points in our analysis. The following plot shows an example of using all data points vs subsampled data to construct the partial dependence and confirms that subsampling does not alter the overall relationship of the variable of interest (here, slope) with response variable (snow depth). We will improve the description of the partial dependence estimation of the variables in the revised manuscript.

[Figure]

ii.  RF model performance (i.e., modeled vs. observed snow depth, current Figure 7) for training and prediction periods (a 2x3 panels plot, with the top row for training, and the bottom row for OOB).

We did not separate our data into training and testing sets, so that we would not create an artificial bias by data splitting. We used all data in random forest models but used the Out-Of-Bag (OOB) validation statistics to assess model performance. So, 2/3 of sample data (in-bag) was used to train the model and the remaining 1/3 (OOB data) was used to estimate how well the trained model performs. However, we acknowledge that the OOB was not sufficiently described in our manuscript. In L252, "a validation set" should be corrected as "a cross validation method". This in-bag and OOB sampling procedure is akin to the much used k-fold cross-validation approach (Probst and Boulesteix, 2017; Tyralis et al., 2019). We will update the revised manuscript for a better clarification of the OOB statistics.

iii.  Results for predictor importance (current Figure 5). How different are these compared to those obtained with the training dataset?

Since we did not split data into training and testing sets, we don't have a separate training dataset. Each tree model of the forest was trained using the in-bag sample data. In random forest model, predictor importance is generally calculated using the OOB sample data, that was not used during the model construction.

**Minor comments**

4.  L25: Do the authors mean "physically-based models"? Note that all hydrological models (even simple bucket-style models) are, to some extent, process-based (see discussions in Hrachowitz and Clark 2017).

Yes, we will change in revised manuscript.

5.  L37: Do you mean vegetation density?

Yes, it is vegetation density, to be changed in the revised version.

6.  L39-40: You might want to read and cite the work of Deems et al. (2013).

Thanks. We will cite this reference here.

7.  L43: Please clarify what you mean with high-resolution. In L43 you say <100 m, but in the following line you say <10 m. Also, I suggest providing references for micro and meso-scales, and reviewing the study of Tedesche et al. (2017).

We will correct this sentence as, "Lidar scanning also typically allows capturing high-resolution micro (<100 m) and mesoscale (100 m–10 km) variability and allows producing high resolution snow depth/cover maps". We will also add references for micro and meso scales.

8.  L48-49: The authors should include other studies that also reported multiscale behavior in snow depth (Helfricht et al. 2014; Clemenzi et al. 2018; Mendoza et al. 2020b). The latter is particularly relevant for this study (and the discussion in L432-433, L445-447), since the authors found 4-m scale break lengths (similar to what is reported here) at the only Andean vegetated site they examined.

Thanks. We will cite these refences in this section.

9.  L50: I think the authors want to say "different combinations of processes". Also, I would precise that you refer to the importance of horizontal resolution, not only for the measurement scale, but also to inform model scales.

Thanks. We will change the "different processes" to "different combination of processes" so it would be clearer. We will emphasis the importance of horizontal resolution and amend the text in relevant sections in the revised manuscript.

10. L62: Please note that process-based models are assemblages of hypotheses about the functioning of hydrological systems. Accordingly, models might be missing processes (e.g., avalanches, blowing snow) that are relevant in particular locations, and hence not all of them are applicable to all conditions (see discussions in Clark et al. 2011).

Thanks. We fully agree with your comment. By, "While process-based models are applicable to a wide range of conditions" we indirectly inferred that these models might not be applicable to all conditions. But we will change the sentence accordingly to make this clearer.

11. Table 1: Was the snow-on flight conducted right after a storm? I think this information is relevant to establish possible connections between your snow depth results and dominant winds.

None of the flights were conducted right after a storm. We will add this information to the table.

12. L182-183: If your aim is to analyze wind effects on snow redistribution, you should filter your data considering (i) wind speeds above a threshold (e.g., 4 m s−1) and (ii) air temperature below 0°C when snow transport by wind is most likely to occur (e.g., Trujillo et al. 2007). I also recommend the authors to revise Li and Pomeroy (1997).

Thanks. We did not intend to carry out a thorough analysis of wind effects on snow redistribution, but rather the impact of topographical, vegetation and forest edge effects on snow depth variability. We think that the winter season wind roses provide a sufficient depiction of the dominant wind speed and directional distribution at each site for this purpose.

13. L197-198: I recommend the authors to explain with words what the canopy cover and the gap fraction are, before providing details on how you compute their values.

Thanks. We will add these details in the revised manuscript.

14. L205-209: I think this explanation would greatly benefit from a diagram showing what a forest edge is, a hypothetical dominant wind direction, windward, leeward, and the maximum search distance.

Thanks. We will add the following diagram in the revised manuscript:

[Figure]

10H indicates the maximum search distance ($d_{max}$) in the open field from the forest edge in windward and leeward direction, 1H indicates the maximum search distance in the forest from the forest edge in the

windward and leeward direction, and 2H indicates the maximum search distance northward of the forest edge. Forest edge boundary was extracted from the site variable.

15.   L241: How did you define the maximum lag distance for variogram calculations? Note that Sun et al. (2006) recommended setting it to half of the maximum point pairs distance for variogram calculations.

We also used half of the maximum point pairs distance in our variogram analysis, which will be mentioned in the revised text.

16.   Figure 3: It would be helpful having the site names here, hopefully between the top and the bottom panels. You could also include the maximum snow depth, the coefficient of variation (CV) and the skewness to compare field vs. forest. Even more, the authors might consider merging Figures 1 and 3 into a unique Figure, to make it easier to see the snow accumulation patterns they describe with the various land cover types.

Thanks. We will update Figure 3 by considering your comments. We will add forest and field margins in the snow depth maps, so it will be easier to grasp the accumulation patterns in forest and field we are discussing in the text.

17.   L279-280: This is really hard to visualize. Do we really need this level of detail?

We wanted to mention that roads and house premises are snow free (zero snow depths) due to snow clearing operations.

18.   L286-287: Where are those snow depth intervals coming from? They don't seem to reflect the actual ranges.

Those ranges are extracted from the mean snow depth values in histograms in Sainte-Marthe and Saint-Maurice. We agree that the sentence did not correctly express what we meant. We will change the sentence as "snow depths in Sainte-Marthe (0.246–0.369 m, forest and field respectively) appear to be lower on average than in Saint-Maurice (0.592–0.600 m, forest and field respectively)".

19.   L295-296: Where are you showing this? I don't see it in Figure 3.

We refer to figure 4, semi-variograms here.

20.   L297: I think what you actually mean is "multi-scaling". Multifractal implies a continuous spectrum of fractal dimensions (Mandelbrot 1988). Also, you should define what a fractal is (perhaps in the methods section).

Thanks. Yes, we meant "multi-scaling". We will correct this in revised manuscript.

21.   Section 3.3: there are too many acronyms in this manuscript, making it difficult to follow the reasoning. I suggest deleting some or replacing them for more intuitive ones.

Thanks. We will address this issue and change accordingly in revised manuscript.

22.   Figure 5: Perhaps it would be easier to understand these results if you linked the different symbols with straight lines.

Thanks. Your comment meets that of referee # 3, we will change figure 5 to a horizontal bar chart in the revised manuscript.

23. L357-358: this is not clear from Figure 6. Can you please provide a better explanation?

We will change it as, "However, contrary to Sainte-Marthe, the overall relationship (considering the field+forest curve) is dominated by intra forest variations in LAI rather than the difference between the field and forest". In contrary to Sainte-Marthe field+forest curve of LAI, Saint-Maurice field+forest LAI curve more closely follows the forest LAI pattern. This indicates that the LAI variation in Saint-Maurice forest have more influence of field+forest LAI curve than that in Sainte-Marthe.

24. L383: I think it's the other way. Figure 7 shows RF model estimates vs. observed snow depth.

Thanks. We will correct this.

25. L424: 'more spatially continuous'. What do you mean with this? It seems to contradict the previous sentence.

We meant a continuous snowpack over the distance of the scale break. For example, the snowpack in Montmorency is more variable (high semi-variance value) and comparatively continuous over the distance of the scale break than the snowpack at the other two sites.

26. L444: I think here you should cite Mendoza et al. (2020b) and NOT Mendoza et al. (2020a).

Thanks. We will correct this in the revised manuscript.

27. I think section 4.3 could be largely condensed. You may also consider shortening the introduction.

Thanks. We will divide the section to sub sections in the revised manuscript.

28. L479: 'At the combined scale'. I think it is more appropriate to write 'At the full domain'.

Thanks. We will change this in revised manuscript.

29. L489: I think that you mean Hydrologic Response Units (HRUs).

Thanks. Yes. We will correct this mistake in revised manuscript.

30. L490: 'successfully modeled'. Can you please provide some numbers demonstrating that the hydrologic modeling was indeed successful?

They used CRHM to model hydrological processes in agro-forested catchment in southern Quebec. They reported a NSE of 0.57 over 23-year simulation of SWE. We will add these details in revised manuscript.

31. L518: These results seem quite poor. Did you compare RF performance with multiple linear regression models?

Yes, we did. Performance of multiple linear models were poorer than the RF performance. We will add a brief comparison of them in the revised version.

32.   L532-533: I would not even mention those references, since NSE is not a good metric to assess the spatial accuracy of model simulations. There are other performance measures for such purpose (Koch et al. 2018; Demirel et al. 2018; Dembélé et al. 2020).

Thanks. We will amend this in the revised manuscript.

33.   L558: I would avoid referring to 'improved accuracy', since RF model results are quite poor.

Still, the inclusion of forest edge metrics does improve the accuracy, even if the overall resulting accuracy remains moderate. We will adjust the wordings in revised manuscript.

34.   L567: 'forest structure variability'. Do you mean spatial or temporal variability?

We meant the spatial variability of the forest structure.

**Suggested edits**

I have provided some editorial suggestions. However, I think that the manuscript would tremendously benefit from a language revision.

Thanks. We will address the following suggestions in the revised manuscript and revise the writing.

35.   L13: 'for the accurate prediction' -> 'for accurate predictions'.

36.   L28: delete 'problematic'.

37.   L34: 'on the downstream hydrograph' -> 'on downstream hydrographs'.

38.   L56: 'The knowledge' -> 'the estimation'.

39.   L60: delete 'modeling approaches like'.

40.   L77: 'that used RF algorithm to express' -> 'quantifying'.

41.   Table 1: 'mm/y' -> 'mm/yr'.

42.   L150: 'were quantified' -> 'were obtained'.

43.   L153-155: I suggest writing the sentence between parentheses in a separate sentence.

44.   L162: 'When taking into account' -> 'Considering'. Delete 'that was typically'.

45.   L163: add a comma after 'environment'.

46.   L165: 'to represent' -> 'represent'.

47.   L165: 'As well': I find this term quite odd. I would replace by 'Additionally', 'Further', 'Moreover', etc. (this comment applies for the entire manuscript).

48. L180: 'which gives' -> ', providing'.

49. L182: 'from the hourly' -> 'from hourly'.

50. L250: 'two thirds… is' -> 'two thirds… are'.

51. L266: delete 'of a variable'.

52. L314: delete 'to allow'.

53. L321-322: rewrite as '…LAI and WFE have the highest (64 %) and least (3 %) impacts, respectively, on snow depth…'

54. L323: 'acting in forests and fields' -> 'in such environments'.

55. L407: 'In our results' -> 'our results show that'.

56. L415: '… suggests microtopographic… ' -> '…suggests that microtopographic…'

57. L458-459: Awkward sentence. Please re-write.

58. L462: 'by the preferential' -> 'by preferential'.

59. L470: Delete 'Whereas'.

60. L475: 'and dominates' -> 'dominating'.

61. L487: 'in the melting' -> 'during the melting'.

62. L488: 'challenge the'. Delete 'the'.

63. L512: 'could be due to' -> 'could be explained by'.

Greenwell, B. M.: pdp: An R package for constructing partial dependence plots, The R Journal, 9:1, 421–436, 2017.

Jerome, H. F.: Greedy function approximation: A gradient boosting machine, The Annals of Statistics, 29, 1189–1232, doi: 10.1214/aos/1013203451, 2001.

Mendoza, P. A., Musselman, K. N., Revuelto, J., Deems, J. S., López-Moreno, J. I., and McPhee, J.: Interannual and seasonal variability of snow depth scaling behavior in a subalpine catchment, Water Resour. Res, 56, e2020WR027343, doi: 10.1029/2020WR027343, 2020.

Probst, P. and Boulesteix, A.-L.: To tune or not to tune the number of trees in random forest, Journal of Machine Learning Research 18, 6673–6690, 2017.

Tyralis, H., Papacharalampous, G., and Langousis, A.: A brief review of random forests for water scientists and practitioners and their recent history in water resources, Water, 11, 910, 2019.

---

## Author Comment (AC2)

We thank the anonymous referee #2 for his/her thorough review with constructive comments and suggestions that certainly will improve the manuscript. In the following, we will address the referees' comments point by point. We mark "black" the comments given by the referee, and our responses in "blue".

**Comment on tc-2022-124**

Anonymous Referee #2

Referee comment on "Topographic and vegetation controls of the spatial distribution of snow depth in agro-forested environments by UAV-lidar" by Vasana Dharmadasa et al., The Cryosphere Discuss., https://doi.org/10.5194/tc-2022-124-RC2, 2022

The study by Dharmadasa et al. explores snow depth distribution at three Canadian sites (agro-forested and boreal forest) based on snow depth maps derived from UAV-LiDAR. Scaling behavior is investigated, and random forest models are used to assess the importance of various topographic and vegetation controls on these snow depth distributions. The topic of the study is of interest to the community, and in general, the manuscript is neatly organized, and the figures are well made. However, I have some methodological concerns that need to be addressed before this manuscript can be considered for publication. A major issue is certainly that the data at hand may not be sufficient to reach the conclusions drawn in the study – in the current state, the key findings and novelties of the study as well as the potential impacts of these findings are not highlighted well enough. Please find my major and minor comments detailed below, as well as some suggestions that I believe would make the study more novel and convincing.

We thank referee #2 for the challenging remarks and appreciate the overall potential he/she sees in our study. We acknowledge that the key findings, potential impacts of the findings and novelties were not highlighted well enough, and we will address this thoroughly in the revised manuscript. We will also take into consideration the reviewer's suggestions in the revised version that will significantly improve the quality of the manuscript.

**Major comments:**

1. In the methods, the authors should state more explicitly that the datasets are published already. Some redundancy with their article published earlier this year is unavoidable but should be kept to a limit. There are instances where the text could be shortened and simply refer to Dharmadasa et al. 2022, I have pointed these out in the minor comments. Watch out for self-plagiarism - Figure 1 and Table 1 are almost identical to Dharmadasa et al. 2022, and need to include a proper reference (e.g. 'adapted from').

   Thanks. We will add "adapted from" for Figure 1 and Table 1 and change the text accordingly when we refer Dharmadasa et al. (2022).

2. I am not convinced by the choice of aggregating the variables to different resolutions (10-20m) for the RF modelling and related analysis, in my opinion this raises some problematic questions:

- Firstly, I don't understand why the aggregation of the vegetation parameters is needed at all (L200ff). 'Canopy height' and 'Tree height' is not necessarily the same thing, and I see no problem with computing canopy height if the pixel size is smaller than the tree crown. Doing so would actually allow extracting the small canopy gaps that have been shown to be the main source of forest snow variability in some other studies (cited in this manuscript), while these gaps are averaged out if the pixels are aggregated to 10-20m resolution. This averaging is likely masking some dependency of snow depth distribution on forest structure.

- Likewise, averaging topographic variables will average out most of the micro-topographic variability that I understood to be the focus of the study. Again, some dependency between snow depth and these variables may be masked by this aggregation. It should be clarified whether the authors are trying to quantify micro-topography or topography.

- I find it particularly problematic to use an aggregation that is larger than the scale break identified in Section 3.2. Doesn't this mean that the variability that you are trying to explain is averaged out?

- Finally, aggregating leaves you with a rather small sample size, as the surveyed areas are quite small. Especially in the case of Montmorency, the field landcover type covers a very limited area only.

- I would find the analysis more convincing if it was conducted at the 1.4 m resolution, I suggest doing that in view of a resubmission.

  The rational for the initial analysis was to study the snow cover heterogeneity at the scale of single-tree. But we agree with your comment that aggregating does average out the variability smaller than the single-tree scale we considered. Considering your comments and suggestion, we repeated the analysis at the 1.4m resolution. Results show an overall improvement of the RF model performances at all sites (field+forest $R^2$ in Sainte-Marthe is 0.66, Saint-Maurice is 0.46 and Montmorency is 0.3 now). The importance of the variables at each site were only slightly changed. At agro-forested sites, windward forest edge effect and microtopography still have the highest impact on snow depth variability and in coniferous site, it is still the forest structure variability (LAI). But windward forest edge effect in forested areas at the two agro-forested sites are now evident compared to the coarse scaled data. We will replace the whole RF analysis by the updated analysis done at 1.4m resolution results in revised manuscript.

3. Some parts of the results chapters require more detail / explanation, and the discussion needs to be more convincing. Some examples:

- Section 3.1: The large overlap of forest and field histogram makes me wonder whether the difference between the two distributions is statistically significant – testing this would be appropriate (see e.g. Currier et al. 2018 for examples of such tests). The inter-site comparison is problematic because data acquisition did occur in different years.

  We will test the statistical significance of field and forest snow depths and include them in the revised version. Initial plan for the study was to acquire data in Montmorency in 2020 as well. But in compliances with the COVID-19 regulations that were in effect during that time, we were not allowed to access the site for surveys. However, the long term ECCC data shows that Montmorency always receives much higher snowfall than the two agro-forested sites. Therefore, we decided to utilize the data available at hand for the analysis.

- Section 3.2: It is unclear how the scale breaks were identified.

  We will update this section in the revised manuscript based on extensive comments made by reviewer #1 on this point. Please see our detailed response regarding the updated variogram analysis along with an explanation of identification of the scale breaks in AC1 report (response 1 to reviewer #1).

- Section 3.3.1: You need to better justify why you computed so many topographic and vegetation variables if most of these remain unused in the analysis / RF model. You also need to explain why you used the same predictors at all sites – for instance, NFE and WFE are homogeneous across the forested area in Montmorency forest so it is not surprising that they have no predictive power.

We discarded Elevation from the analysis since the elevation difference at any of the three sites was too small (Table 1) to produce any meaningful local orographic effect on precipitation, or adiabatic effects on air temperature, e.g., Mazzotti et al. (2019), and could mask other local topographic effects on accumulation related to slope, aspect and terrain roughness (wind sheltering), due to collinearity. And, irrespective of the variable type, we excluded collinear variables prior to building the RF models using the variance inflation factor (VIF) function in R.

Vegetation descriptors (LAI, CC, GF, and CH) were strongly intercorrelated (with r of 0.84–0.97) and could not be used together in a predictive model (at least not without compromising the interpretation of variable importance in the RF model).Therefore, LAI was selected to use in RF analysis as it has been shown to be a strong predictor of snow accumulation (Hedstrom and Pomeroy, 1998; Broxton et al., 2015). In the revised manuscript we added a new sensitivity analysis which shows that the choice of forest structure descriptor has a negligible impact on the performance ($R^2$) of the model.

In the original manuscript we used the same variables at all sites for the RF analysis to allow for an easier intercomparison of the effect of driving variables between the sites. In the updated analysis (RF analysis with 1.4m grids), we decided to select the forest edge matrix guided by the landscape setting at each site. E.g., in Montmorency, we now include the leeward forest edge (LFE) matrix instead of windward forest edge (WFE) matrix because the leeward edge seemingly has more influence on snow depth variability with its larger extent than the windward edge (as shown in following figure). This is also more logical as the open areas in Montmorency constitutes a large gap within an overall forested environment, so deposition is expected leeward of the forest edge with little remobilization (erosion) within the gap. It is also confirmed by the improved $R^2$ of 0.56 in RF field in Montmorency. Also, at 1.4m scale, windward, leeward and northward forest edge pixels in the forest are more evident and variable than in the previously used coarse scale at all sites. The variables used in the updated RF analysis in Montmorency now include *Slope, Aspect_WE, Aspect_SN, TWSI, LAI, LFE, and NFE.*

[Figure]

Montmorency variables

In Sainte-Marthe, both WFE and LFE have large extents (following figure) but are collinear due to the two dominant and opposed wind directions. Including both variables in the RF model would thus compromise the interpretation of the variable importance; collinearity generally reportedly makes it difficult to separately evaluate the predictive power (variable importance) of the predictors (Bair et al., 2018). Hence, we opted to use the WFE only in the final RF analysis and will now discuss this choice on the interpretation of forest edge effect at Sainte-Marthe. The variables used in the RF analysis at Sainte-Marthe are *Slope, Aspect_WE, Aspect_SN, TWSI, LAI, WFE, and NFE.*

[Figure]

Sainte-Marthe variables

In Saint-Maurice, LFE has only a few pixels (following figure) and was hence omitted from the RF analysis. The variables used in the RF analysis in Saint-Maurice are *Slope, Aspect_WE, Aspect_SN, TWSI, LAI, WFE, and NFE.* We will improve section 3.3.1. by better explaining the selection of variables at each site in the revised version.

[Figure]

Saint-Maurice variables

- Section 3.3.3: This section is a bit lengthy, and the key messages don't quite come through. Maybe it would be better to show fewer subpanels in Figure 6 and focus on the variables that actually exhibit interesting relationships to snow depth.

  Thanks. We will revise this section.

- In the discussion, you need to comment on the actual benefit of such an RF approach. The model performance shown in 3.3.4 is pretty poor, and it's not straightforward to extract the interesting information from the plots in Figure 6. In fact, I felt like the most insightful Figure to grasp the snow depth variability was Fig 3, where increased accumulation in sheltered locations is visible by eye.

  RF notably allows capturing non-linear relationships between snow accumulation and landscape variables. We will discuss the benefits of using RF approach in the updated discussion section, notably compared to traditional multiple linear regression models.

4. Given that the datasets have already been presented and used in an earlier study by the same work, the added value of the analysis presented in this manuscript seems a bit limited and the novelty of the study is not emphasized enough. What are the key findings, and where will they be useful? I realize that it's not easy to get more out of such a limited amount of data, but I tried to make sume suggestions that the authors could consider in view of a re-submission.

The earlier study solely focused on the accuracy of the UAV-LIDAR system, a necessary methodological step, and did not analyze the spatial heterogeneity of the derived snow depth maps. We thank the referee #2 for his/her insightful suggestions to improve the quality of our manuscript. We will adopt the suggestions in the revised manuscript as much as possible.

- If the authors have the opportunity to acquire more data in the upcoming winter, the authors should consider postponing the final analysis to after the upcoming season. Repeated flights over the same site would allow for a much more insightful analysis. It is very difficult to draw conclusions on individual processes based on snow distribution data from one acquisition only, which I think is part of the reason why the discussion does not seem very conclusive. For instance, it could be interesting to see if the snow depth maxima at the forest edges are a recurring feature, and if their effect persists throughout ablation (that's just an idea, but one could do much more). It would also be good to survey all sites in the same year to allow a more convincing comparison between sites.

We agree that having repeated flights in the same areas in several winters would be better. But unfortunately, due to logistical reasons, we will not have the opportunity to acquire more data in the upcoming winter in these areas. This caveat will be mentioned in the discussion. However, a new, ongoing study will explore the temporal variability of the snow depth in a coniferous site. Still the analysis presented here is thought to largely reflect the typical conditions at the sites and to portray key differences between agro-forested environments and boreal forests. To place the studied years in climatological context, we will compare the climatological conditions in the study year (snowfall, air temperature, wind speed, and wind direction) with historical conditions at the sites and include in the revised version.

- Since the authors analyzed many more terrain and vegetation variables than they ended up using in the RF model, it could be interesting to dedicate a section on the physiographic variables themselves, attempting to identify a set of variables useful to characterize this sort of landscape. Maybe in comparison to variables that have been related to snow distribution in other studies. For example: Elevation has been found to exert a main control on snow depth in complex terrain in other studies, but it is not a really 'useful' predictor at the sites used in this study – is this a consequence of the site choice, or is this site representative of the terrain found in the whole ecoregion?

One particularity of our sites (also related to the scale of the analysis) is the negligible elevation range. Many studies conducted in mountainous environments have shown the preponderant influence of elevation on the distribution of snow cover. While the elevation range becomes important over larger extent on the Canadian shield (Montmorency-type physiography), the low elevation St-Lawrence lowlands (Sainte-Marthe and Saint-Maurice) remain mostly flat and local topography (terrain roughness) and land cover and land use are expected to control the spatial distribution of the snow cover. We will add a section in the discussion in the revised version to discuss the potential predictors at similar landscapes.

- Exploring the relationships between snow and physiographic variables at different spatial aggregation levels could be interesting.

Thanks. The updated manuscript will replace the initial aggregated scale by the high-resolution 1.4m analysis. We plan however to include a section in the revised paper to compare the RF results at 1.4m scale to single-tree scale results (coarse scale we used earlier).

- Applying additional modelling approaches (statistical, or even physically based) to compare with the RF model could be insightful, especially to draw conclusion on the utility of these findings for later work or practical applications.

  Thanks. We will compare the RF results with multiple linear regression at 1.4m scale and add them in the revised version.

- Adding an application of the model results would be a nice addition – e.g. suggest tiling approaches, extend to larger area or entire watershed, etc.

  Thanks. We agree that this will be a great addition to our study. However, we anticipate that the revised manuscript will have two or more pages in addition to the current 29 pages after including new sections and plots. As much as we appreciate this suggestion, we think addition of this suggestion would make the revised paper lengthier. Moreover, we think we will be able to highlight the novelty of the study by implementing the suggestions 2,3, and 4.

**Minor comments** (including wording/language suggestions)

L37 'topography and vegetation type, and density' -> you mean vegetation density? Sentence doesn't read very well, consider rephrasing

Yes. It is vegetation density. We will rephrase this sentence.

L46: a very nice and comprehensive paper on the topic: https://doi.org/10.1029/2011WR010745 - I suggest including this reference

Thanks. We will include this reference.

L51 'a short scale break is reflected by interception' -> I would say it's the other way round?

We meant that interception is causing the short scale break. We will change the sentence as "For instance, these studies emphasized that canopy interception causes a short scale break distance in forested areas (9– 12 m) where the effect of wind redistribution is minimal".

L62: You should refer to much more recent developments of process-based models, as some of these models now actually do resolve small-scale variability due to heterogeneous canopy structure. See Broxton et al. 2015 (already cited elsewhere) and Mazzotti et al. (https://doi.org/10.1029/2019WR026129 and https://doi.org/10.1029/2020WR027572).

Thanks. We will refer to recent developments of process-based models and amend the text accordingly in the revised version.

L93: 'one of the earliest results […]'. I think this is not a very fair 'selling argument' for this study, since the data used for the analysis has already been presented in another paper.

Thanks. We will remove this sentence.

L108: incomplete sentence (WMO's station network?)

Thanks. It is WMO's station network. We will correct this.

Table 1: Winter season -> snow cover period?

Yes.

L136 is this vertical or horizontal accuracy, or both?

It is the vertical accuracy.

L164: what do you mean by 'multipath effect'?

Internal reflection of GNSS signals against obstacles (trees) before reaching the receiver.

L165: you just said the accuracy is comparable to previous studies, so what is the improvement? I would omit the entire end of the paragraph from 162 onward and just refer to Dharmadasa 2022.

We will remove the section from L162 onwards and refer to Dharmadasa et al. (2022).

Section 2.2.2-2.2.4: Please specify that maps of the variables are found in the supplementary material, I was missing those maps here and found them only much later. Note that the figures in the supplement should include the units for all variables.

Thanks. We will correct this in the relevant sections and add the units in supplement figures.

L197-199 Is GC = 1-CC? and at what resolution are these metrics calculated, also 1.4m?

Yes. They were also calculated as the same size as the LAI. We will add this in the revised version.

L201: How did you estimate crown diameter?

We calculated the crown diameter in LiDAR360 software. LiDAR360 uses Li et al. (2012) point cloud segmentation algorithm to segment individual trees. Crown diameter is one of the output we get from this segmented trees in the software (Greenvalley-International, 2020). We will describe this in the revised manuscript.

L224: This approach seems a combination of Currier & Lundquist and the DCE presented by Mazzotti et al 2019 (which however has no notion of search distance contrary to your method). Maybe worth noting?

Thanks. We will include this in the revised version.

L256: It is not very clear how you define the variable 'Site', or at least it wasn't to me when looking at the descriptor maps in the supplementary material and comparing with the other vegetation metrics. I think this is quite crucial for understanding the edge metrics, hence I more detail is needed here.

The binary variable, site was derived according to the field and forest area boundaries manually mapped at each study area. If there is a forest patch in the field, we considered that patch as forest and vice versa. For instance, in Sainte-Marthe we considered forest patch located to the southwest in field as forest in addition to the large forested area. But in Saint-Maurice and Montmorency, derivation of site variable was more

straightforward as field and forest patches were well separated and so easily distinguishable than in Sainte-Marthe (i.e., there were no forest patches in field as in Sainte-Marthe). Once we delineate the forest and field boundaries, we assign 0 to field and 1 to forest. We will add more details to describe the derivation of site variable in the revised version.

L264 Tenses are inconsistent

Thanks. We will correct this.

L266: Variable importance of a variable? Consider rephrasing

Thanks. We will change this as "importance of a variable".

Figure 3: specify whether you used the binary variable or the land cover classification to create the histograms.

We used the boundaries from site variable to create the histograms. Figure 3 will be updated in the revised manuscript indicating the forest and field boundaries.

L308: how did you come to this conclusion?

Depending on the correlation coefficient value. Please note that in the updated analysis at 1.4m grid resolution, we now included the leeward forest edge (LFE) variable in Montmorency RF model, since it has more influence on the snow depth predictions than windward forest edge variable (which was also confirmed from the larger extent of LFE raster than WFE).

L310 'collinearity analysis suggested discarding GF and CH in favor of LAI at the two agro-forested sites, while LAI was instead flagged as colinear instead of GF and CH in the coniferous site'. Please rephrase – 'LAI was flagged as colinear' is unclear (colinear to what?)

Thanks. We will correct this in the revised version.

Figure 6: Y-axis label missing (snow depth)

Thanks. We will correct this in the revised version.

L445-446: Unloading through branches should reduce spatial variability, no?

Yes. It could. Lower semi-variance value in temperate forests compared to coniferous forest in figure 4 suggests that the overall spatial variability of the snowpack is less in these forests compared to that in coniferous forest. Yet, this could still result in a smaller correlation length in snow depth because of the interwind nature of branches on deciduous trees.

L511-513: The counterintuitive […] stations at the site. -> I don't understand what you are trying to say here.

We wanted to say, in contrast to preferential deposition on northerly slopes in the northern hemisphere, Montmorency field accumulates more snow on southerly slopes. This could be due to the influence of the various meteorological stations at the site. We will change the sentence in the revised version.

L565ff: this section needs to acknlowledge hyper resolution process based (physically based) models (see earlier comment). There are ways to account for fine scale canopy structure, while I would say that the terrain roughness still represents a major difficulty.

Thanks. We will address this in the revised version.

L483: I think this should be 'Hydrologic response units'

Yes. Thanks. We will correct this in the revised version.

Section 4.3: The work from Safa et al. (https://doi.org/10.1029/2020WR027522) needs to be included in this discussion – they applied RF models as well.

Thanks. We will add this reference in the revised version.

Bair, E. H., Abreu Calfa, A., Rittger, K., and Dozier, J.: Using machine learning for real-time estimates of snow water equivalent in the watersheds of Afghanistan, The Cryosphere, 12, 1579–1594, doi: 10.5194/tc-12-1579-2018, 2018.

Broxton, P. D., Harpold, A. A., Biederman, J. A., Troch, P. A., Molotch, N. P., and Brooks, P. D.: Quantifying the effects of vegetation structure on snow accumulation and ablation in mixed-conifer forests, Ecohydrology, 8, 1073–1094, 2015.

Dharmadasa, V., Kinnard, C., and Baraër, M.: An accuracy assessment of snow depth measurements in agro-forested environments by UAV lidar, Remote Sensing, 14, 1649, doi: 10.3390/rs14071649, 2022.

GreenValley-International: LiDAR360 User Guide, GreenValley International, Ltd, Berkeley, CA, USA, 2020.

Hedstrom, N. R. and Pomeroy, J. W.: Measurements and modelling of snow interception in the boreal forest, Hydrological Processes, 12, 1611–1625, 1998.

Li, W., Guo, Q., Jakubowski, M. K., and Kelly, M.: A new method for segmenting individual trees from the lidar point cloud, Photogrammetric Engineering & Remote Sensing, 78(1), 75–84, 2012.

Mazzotti, G., Currier, W., Deems, J. S., Pflug, J. M., Lundquist, J. D., and Jonas, T.: Revisiting snow cover variability and canopy structure within forest stands: Insights from airborne lidar data, Water Resour. Res, 55, 6198–6216, doi: 10.1029/2019WR024898, 2019.

---

## Author Comment (AC3)

We thank the anonymous referee #3 for his/her thorough review with constructive comments and suggestions that certainly will improve the manuscript. In the following, we will address the referees' comments point by point. We mark "black" the comments given by the referee, and our responses in "blue".

Comment on tc-2022-124 Anonymous Referee #3 Referee comment on "Topographic and vegetation controls of the spatial distribution of snow depth in agro-forested environments by UAV-lidar" by Vasana Dharmadasa et al., The Cryosphere Discuss., https://doi.org/10.5194/tc-2022-124-RC3, 2022

**Major comments:**

1. This manuscript is using UAV-lidar data to understand the snow-depth heterogeneity in agro-forested environments and boreal forests. Since the author also used the slope and aspect as topographical variables for studying the snow-depth distribution. However, at site Saint-Marthe and Saint-Maurice, the elevation difference is actually very small comparing to the spatial scale of the study area, it would be kind of surprising to observe any meaning effects from topographical variables in these 2 study areas. And although Montmorency has a elevation difference of 20 meters in the study area, it seems the main area is facing towards south east direction. It would also be helpful to visualize the distribution of the slope and aspect of the 3 study areas to make sure there is significant variability in these predictors of snow-depth before feeding them into a model like random forest.
   Thanks. Please kindly note that we included the plots of the independent variables in supplementary material.

2. It is not very clear on how the forest edge descriptors are derived, it would be helpful to have a visualized illustration to demonstrate how variables are derived based on canopy cover data from lidar.
   Thank you for the suggestion. We will include the following visualized illustration to demonstrate how forest edge descriptors were derived in the revised manuscript.

[Figure]

10H indicates the maximum search distance ($d_{max}$) in the open field from the forest edge in windward and leeward direction, 1H indicates the maximum search distance in the forest from the forest edge in the windward and leeward direction, and 2H indicates the maximum search distance northward of the forest edge. Forest edge boundary was extracted from the site variable.

3. The 2nd objective raised in the Introduction part for this manuscript seems to be extremely open ended. Is there a particular hypothesis the authors would like to test with the used dataset and validate the hypothesis throughout the manuscript? The current objective of "exploring the relationship between snow depth, topography, and forest structure" seems too vague and not specific enough.

Thank you for point this out. We will make the second objective more specific in the revised manuscript. We will add the following sentences at the end of this paragraph. "Forest edge effects are expected to exert a greater control on snow depth variability in agro-forested sites and the forest structure is expected to be dominating the snow depth variability in coniferous forested environment."

**Minor comments;**

Figure 1 – it looks like the first 2 study areas are very flat with low elevation and the 3rd one has elevation difference and the elevation is much higher. It the precipitation in this area affected by orographic effect as well?

At the scale we considered for our study, elevation difference at any of the sites were too small to produce any meaningful orographic effects. Even at larger scales, the first two sites are generally flat, hence will not have any orographic effects on precipitation. But in the third site, Montmorency, when the scale of the study area becomes larger (beyond that analyzed here), orographic effects would be expected to have an impact on precipitation.

Line 154-155: given it is 1 m diagonal cross shape, why it is 1.4x1.4 m grid cell? Isn't it going to be 0.7m x 0.7m grid cell instead?

The manual measurement strategy with 5 measurements at each sampling location we adapted was as follows:

[Figure]

Hence, the length in one side of the grid cell is $(1^2+1^2)^{1/2}$. Which is 1.4m.

Line 162-165: it is not clear why UAV-lidar is more robust and the technology represents an improvement to previous studies.

By considering your comment and a similar comment from referee #1, we will rephrase and remove some parts from this section in the revised manuscript.

Line 184-186: The closest weather station is 19 km away from Saint-Maurice. Is the wind data going to be trustable for this site given it is very far and the wind speed and direction can be quite different comparing to the actual on the site, right?

This is the closet wind station available for this site. Given that the flat topography of this area, wind speed is expected to be not driven by topography and spatially coherent over large scales.

Line 193: how is LAI, CC, and GF calculated? By using Lidar360 software?
Yes. They were calculated in Lidar360 software. This will be better explained in the revised version.

Line 202: why the grid size for vegetation is so much larger than the resolution of the snowdepth (1.4x1.4 m). It seems the vegetation grid resoluation is so much coarser and are we able to capture all the forest variable based on such a low resolution?
The initial goal was to analyse snow depth variability at a scale of single-tree level. We selected the resolutions in a way that at least one tree will be inside the grid cell. However, we agree that the variability averages out inside a coarse grid cell. Based on your comment and that of referee #2 we redid all analyses at the 1.4m resolution. Random forest analysis results and discussion sections will be replaced accordingly in the revised manuscript.

Section 2.2.4, please see major comment #2, it is not very clear how d and d_max are derived based on the forest-covered lidar data.
We will add an illustration to better explain the derivation of d and d_max.

Line 250-253, are hyperparameters in Random Forest tuned or selected before training each model?
Hyperparameters in random forest at each site were tuned before training each RF model. This will be mentioned in the revised version.

Line 257: It is not very clear how forested vs. fields are defined. It would be helpful to have a map of these site showing this binary variable. And why don't we use this binary variable directly in the RF model directly? Is it to show at different area how other variables affecting snow-depth differently?
The binary site variable was derived according to the field and forest area boundaries manually mapped at each study area. If there was a forest patch in the field, we considered that patch as forest and vice versa. For instance, in Sainte-Marthe we considered forest patch located to the southwest in the field as forest in addition to the large forested area. In Saint-Maurice and Montmorency the derivation of the site variable was more straightforward as field and forest patches were well separated and so more distinguishable than in Sainte-Marthe (i.e., there were no forest patches in field as in Sainte-Marthe). Once we delineate the forest and field boundary, we assign 0 to field and 1 to forest. We will add more details to describe the derivation of the site variable in the revised version.

Please note that the plot of site variable at each site was included as supplement figures.

Figure 3: it is a bit surprised to me at Montmorency there is not many data points for under canopy. Then we might not be able to observe a lot of under canopy snow-depth signals.
Since Montmorency has a thick evergreen canopy cover, we did not get ground returns under the canopy at some locations. Hence, we lacked the snow depths under canopy on such occasions. This which is a limitation of the UAV-LIDAR system in thick coniferous covers is acknowledged in section 4.4 Limitations of the study.

Figure 4: how is the scale break selected? Please describe that in the Method section.
We will include this in the revised version of the manuscript. This comment resonates with other referee's comments regarding the scale break identification. For the time being, we provide a detailed description of this in response (response 1) to referee #1's first extensive comment on this point.

Figure 5: it might be better to use bar chart with different colors. It is a bit difficult to differentiate color the marker styles on this scatter plot.
Thanks. We will update figure 5 to a horizontal bar chart in the revised manuscript.

Section 3.3.3: there is only one line in the Method section (line 267) discussed about the partial relationships of preditor variables with snow depth. It is still not very how that is calculated. Please add details in the Method section.

Thanks. We will integrate this in the method section in the revised version.

Figure 7, it looks like the model is not performing very well at Saint-Maurice and Montmorency. The slope of the predictioned vs. observed is not close to 1. What would be the reason that the trained RF model is underfitting and has this systematic bias?

The performance improved with the higher (1.4m) resolution model, but the poorer performance at Saint-Maurice and Montmorency could be due to underlying processes/variables that were not considered in our model, as well as unexplained snow depth variability that is within the system (UAV-LIDAR) detection limit. We will add this issue to the updated discussion.

---

## Author Response (AR1)

**Response to Anonymous Referee #1**

We thank the anonymous referee #1 for his/her thorough review with constructive comments and suggestions that certainly will improve the manuscript. In the following, we will address the referees' comments point by point. We mark "black" the comments given by the referee, and our responses in "blue". We also included the line numbers in revised manuscript in our responses.

**Comment on tc-2022-124**

Anonymous Referee #1

Referee comment on "Topographic and vegetation controls of the spatial distribution of snow depth in agro-forested environments by UAV-lidar" by Vasana Dharmadasa et al., The Cryosphere Discuss., https://doi.org/10.5194/tc-2022-124-RC1, 2022

**Summary**

In this paper, the authors analyze the scaling properties of lidar-derived snow depth and possible dependencies with topographic and vegetation descriptors in two agro-forested and one coniferous site in eastern Canada. They conduct variogram analyses on snow depth fields to find possible scale break lengths that define regions with self-similar behavior, and develop random forest models to characterize predictor importance. The results show scale breaks spanning 4-7 m in forested sites, and relatively longer values in field areas – up to 18 m in wind exposed fields – in agreement with previous studies. The results also show that wind-related forest edge descriptors mostly explain snow depth variability in agro-forested sites, while canopy characteristics (i.e., forest structure) are more important in the coniferous site.

Overall, the topic, research questions and experimental setup are interesting for the snow hydrology community. The literature review and discussions are quite extensive (maybe more than needed), and the graphics included in the manuscript are very nice. I have three major comments that I think the authors should address before this paper is considered for publication. Additionally, the authors will find a set of minor comments and editorial suggestions that may be helpful to improve the quality of this manuscript.

We thank referee #1 for the remarks and overall positive assessment.

**Major comments**

1. Fractal analysis:

i. It is not clear from Figure 4 that scale breaks actually exist and, therefore, all the remaining analyses and interpretations remain unsupported, unless the authors demonstrate quantitatively that the snow depth scaling behavior changes. I recommend the authors to revise Mendoza et al. (2020a) as a reference on how to detect scale breaks in variogram analysis.

We acknowledge this matter that also resonates in other reviewer's comments, that we need to quantitively demonstrate how the scale breaks were identified. Therefore, we repeated the variogram analyses following the method described in Mendoza et al. (2020), as suggested.

ii. The authors need to show quantitatively that variograms hold a power law (which is required to indicate that a spatial pattern is actually fractal). I think that, at the very least, the authors should demonstrate that a linear model in the log-log space holds before and after the scale break, with a high coefficient of

determination (e.g., R2 ï, ³ 0.9). I also recommend the authors to test whether other geostatistical models are more suitable for this data (e.g., spherical, Gaussian).

We repeated the variogram analysis of snow depths at all sites by considering the reviewer's recommendation. The following steps were performed to identify scale breaks:

- First, a change point analysis was conducted on the semi-variograms in log-log space using the ecp package in R (James and Matteson, 2014) to identify possible break points, which allows delineating sections of the semi-variogram with similar trends.
- Then, linear least square regression models were fitted in log-log space for each cluster identified in step 1.
- Finally, we checked whether the changes in the slopes of the log-log linear models were larger than 20 % and that the 95 % confidence limits of the slopes did not overlap, and verified that the $R^2$ was greater than 0.9. If all these conditions were fulfilled, the existence of a scale break was confirmed.

We added this information in 2.3.1 Spatial correlation analysis now (L261-L269).

We replotted the variograms with the updated scale break distances (some of the scale break distances were slightly changed) and linear regressions (Figure 5 in revised manuscript).

iii.   The authors might consider comparing omnidirectional variograms of snow depth (and potential scale breaks) with those obtained from bare earth topography and topography+trees (e.g., Deems et al. 2006; Trujillo et al. 2007). Additionally, the analyses could be enriched by computing directional variograms and associated scaling parameters, in order to establish possible connections with dominant wind directions (e.g., Deems et al. 2006; Schirmer and Lehning 2011; Clemenzi et al. 2018; Mendoza et al. 2020a).

Considering the referee's comment, we compared the omnidirectional variograms of the snow depths with those obtained from bare earth topography and topography+vegetation (Figure 5 in revised manuscript). Generally, it is noticeable that the scale break distances in the forested areas are in the same order of that in topography+vegetation variograms. Scale break distances in bare earth topography are altered by snow accumulation. These observations were now discussed in the revised manuscript accordingly (L334-L344, L467-L515). Additionally, we computed the directional variograms and discussed the isotropic vs anisotropic behavior of snow depth in field and forested areas in the revised manuscript (L351-L356, L516-L528). A new plot with directional variograms was added to the revised manuscript (Figure 6).

We think these amendments and additions will significantly improve the quality of the spatial correlation analysis section of the paper.

2.   Partial relationships: I think the authors should be more quantitative, since the reasoning provided in section 3.3.3 is subjective and difficult to follow. You can easily improve this section by computing the Spearman rank correlation coefficient, and reporting the p-values. I suggest avoiding statements like 'strong relationships', 'stronger that', 'slight decrease with increasing', etc.; instead, you can show the numbers and let the readers judge.

Thank you for this suggestion. We calculated the Spearman rank correlation coefficient and updated all the plots (Figure 8) and changed the discussion accordingly in the revised paper (L401-L421).

3.   The authors may consider using the following sequence to display RF results:

i.    Exploratory analyses with scatter plots of snow depth vs. predictors (current Figure 6). In any case, I recommend the authors verifying these results, since it's very odd that there is practically no scatter along the y axis. Are you displaying all points within your domains in each panel?

Figure 6 (Figure 8 in revised manuscript) displayed partial dependence plots of snow depth against each independent variable, and not simple bivariate scatter plots (which would scatter along the y axis). Partial dependence functions are typically used to help interpret models produced by machine learning models such as random forest analysis (Jerome, 2001). It is a risk adjusted alternative to variable dependence. We used the pdp package in R to obtain the partial dependence of the variables in random forest models (Greenwell, 2017). Here, each partial plot was generated by integrating out the effects of all variables beside the covariate of interest. Partial dependence data in each plot were constructed by selecting points evenly spaced along the distribution of the variable of interest. This subsampling helps to cut down computational time substantially. We used the default subsampling of 51 points in our analysis. The following plot shows an example of using all data points vs subsampled data to construct the partial dependence and confirms that subsampling does not alter the overall relationship of the variable of interest (here, slope) with response variable (snow depth). We also improved the description of the partial dependence estimation of the variables in the revised manuscript (L288-L293).

[Figure]

ii.    RF model performance (i.e., modeled vs. observed snow depth, current Figure 7) for training and prediction periods (a 2x3 panels plot, with the top row for training, and the bottom row for OOB).

We did not separate our data into training and testing sets, so that we would not create an artificial bias by data splitting. We used all data in random forest models but used the Out-Of-Bag (OOB) statistics to assess model performance. So, 2/3 of sample data (in-bag) was used to train the model and the remaining 1/3 (OOB data) was used to estimate how well the trained model performs. This in-bag and OOB sampling procedure is akin to the much used k-fold cross-validation approach (Probst and Boulesteix, 2017; Tyralis et al., 2019). However, we acknowledge that the OOB was not sufficiently described in our manuscript. We updated the revised manuscript for a better clarification of the OOB statistics (L271-L279).

iii.    Results for predictor importance (current Figure 5). How different are these compared to those obtained with the training dataset?

Since we did not split data into training and testing sets, we don't have a separate training dataset. Each tree model of the forest was trained using the in-bag sample data. In random forest model, predictor importance is generally calculated using the OOB sample data, that was not used during the model construction. The

algorithm estimates the predictor importance by estimating how much the prediction error increases when OOB data for the respective variable is permuted while all others are left unchanged (Liaw and Wiener, 2002). (L277-L279)

**Minor comments**

4.  L25: Do the authors mean "physically-based models"? Note that all hydrological models (even simple bucket-style models) are, to some extent, process-based (see discussions in Hrachowitz and Clark 2017).

Yes, we changed in revised manuscript (L25).

5.  L37: Do you mean vegetation density?

Yes, it is vegetation density, changed in the revised version (L37).

6.  L39-40: You might want to read and cite the work of Deems et al. (2013).

Thanks. We now cited this reference here (L41).

7.  L43: Please clarify what you mean with high-resolution. In L43 you say <100 m, but in the following line you say <10 m. Also, I suggest providing references for micro and meso-scales, and reviewing the study of Tedesche et al. (2017).

We corrected and changed this sentence as, "Lidar scanning also typically allows capturing high-resolution micro variability and allows producing high resolution (<10 m) snow depth/cover maps (e.g., Deems et al., 2013; Harder et al., 2020; Koutantou et al., 2021; Dharmadasa et al., 2022)" (L42-L44).

8.  L48-49: The authors should include other studies that also reported multiscale behavior in snow depth (Helfricht et al. 2014; Clemenzi et al. 2018; Mendoza et al. 2020b). The latter is particularly relevant for this study (and the discussion in L432-433, L445-447), since the authors found 4-m scale break lengths (similar to what is reported here) at the only Andean vegetated site they examined.

Thanks. We added these refences in this section now (L48-L50).

9.  L50: I think the authors want to say "different combinations of processes". Also, I would precise that you refer to the importance of horizontal resolution, not only for the measurement scale, but also to inform model scales.

Thanks. We changed the "different processes" to "different combination of processes" so it would be clearer (L50). We changed the last sentence of this paragraph as, "The estimation of this scale break location is important when choosing the horizontal resolution required for remotely sensed or in situ data collection efforts, and model scales in order to represent the snowpack variability at different scales" (L56-L58).

10. L62: Please note that process-based models are assemblages of hypotheses about the functioning of hydrological systems. Accordingly, models might be missing processes (e.g., avalanches, blowing snow) that are relevant in particular locations, and hence not all of them are applicable to all conditions (see discussions in Clark et al. 2011).

Thanks. We fully agree with your comment. By, "While process-based models are applicable to a wide range of conditions" we inferred that these models might not be applicable to all conditions (L63).

11.    Table 1: Was the snow-on flight conducted right after a storm? I think this information is relevant to establish possible connections between your snow depth results and dominant winds.

None of the flights were conducted right after a storm. We now added this information to the table 1.

12.    L182-183: If your aim is to analyze wind effects on snow redistribution, you should filter your data considering (i) wind speeds above a threshold (e.g., 4 m s−1) and (ii) air temperature below 0°C when snow transport by wind is most likely to occur (e.g., Trujillo et al. 2007). I also recommend the authors to revise Li and Pomeroy (1997).

Thanks. We did not intend to carry out a thorough analysis of wind effects on snow redistribution, but rather the impact of topographical, vegetation and forest edge effects on snow depth variability. We think that the winter season wind roses provide a sufficient depiction of the dominant wind speed and directional distribution at each site for this purpose.

13.    L197-198: I recommend the authors to explain with words what the canopy cover and the gap fraction are, before providing details on how you compute their values.

Thanks. We now added these details in the revised manuscript (L193-L196).

14.    L205-209: I think this explanation would greatly benefit from a diagram showing what a forest edge is, a hypothetical dominant wind direction, windward, leeward, and the maximum search distance.

Thanks. We added the following diagram (Figure 3) in the revised manuscript:

[Figure]

10H indicates the maximum search distance ($d_{max}$) in the open field from the forest edge in windward and leeward direction, 1H indicates the maximum search distance in the forest from the forest edge in the windward and leeward direction, and 2H indicates the maximum search distance northward of the forest edge. Forest edge boundary was extracted from the site variable.

15.    L241: How did you define the maximum lag distance for variogram calculations? Note that Sun et al. (2006) recommended setting it to half of the maximum point pairs distance for variogram calculations.

We also used half of the maximum point pairs distance in our variogram analysis. We now mentioned this in the text as, "Half of the maximum point pairs distance (Sun et al., 2006) was taken as the maximum lag distance for the semi-variogram calculations with 50 log-width bins" (L255).

16.    Figure 3: It would be helpful having the site names here, hopefully between the top and the bottom panels. You could also include the maximum snow depth, the coefficient of variation (CV) and the skewness to compare field vs. forest. Even more, the authors might consider merging Figures 1 and 3 into a unique Figure, to make it easier to see the snow accumulation patterns they describe with the various land cover types.

Thanks. We updated Figure 3 (now Figure 4) by considering your comments. We added forest and field margins in the snow depth maps, so it is easier to grasp the accumulation patterns in forest and field we are discussing in the text.

17.    L279-280: This is really hard to visualize. Do we really need this level of detail?

We wanted to mention that roads and house premises are snow free (zero snow depths) due to snow clearing operations.

18.    L286-287: Where are those snow depth intervals coming from? They don't seem to reflect the actual ranges.

Those ranges are extracted from the mean snow depth values in histograms in Sainte-Marthe and Saint-Maurice. We agree that the sentence did not correctly express what we meant. We changed the sentence as "Although the maximum snow depth is higher in Sainte-Marthe (1.8 m) compared to Saint-Maurice (1.6 m), snow depths in Sainte-Marthe are lower on average (mean forest = 0.250 m; mean field = 0.374 m) than in Saint-Maurice (mean forest = 0.591 m; mean field = 0.600 m)" (L323-L325).

19.    L295-296: Where are you showing this? I don't see it in Figure 3.

We refer to the semi-variograms figure (now Figure 5) here (L337-L338).

20.    L297: I think what you actually mean is "multi-scaling". Multifractal implies a continuous spectrum of fractal dimensions (Mandelbrot 1988). Also, you should define what a fractal is (perhaps in the methods section).

Thanks. Yes, we meant "multi-scaling". We corrected this in revised manuscript (L340).

21.    Section 3.3: there are too many acronyms in this manuscript, making it difficult to follow the reasoning. I suggest deleting some or replacing them for more intuitive ones.

Thanks. We addressed this issue in revised manuscript.

22.    Figure 5: Perhaps it would be easier to understand these results if you linked the different symbols with straight lines.

Thanks. Your comment meets that of referee #3, we linked symbols in figure 5 (now Figure 7) with straight lines in the revised manuscript.

23. L357-358: this is not clear from Figure 6. Can you please provide a better explanation?

We wanted to say, "However, contrary to Sainte-Marthe, the overall relationship (considering the field+forest curve) is dominated by intra forest variations in LAI rather than the difference between the field and forest". In contrary to Sainte-Marthe field+forest curve of LAI, Saint-Maurice field+forest LAI curve more closely follows the forest LAI pattern. This indicates that the LAI variation in Saint-Maurice forest have more influence of field+forest LAI curve than that in Sainte-Marthe.

However, we redid the RF analysis at 1.4 m resolution considering referee #2 comments and included in the revised paper. At this scale, this statement is no longer valid, hence removed.

24. L383: I think it's the other way. Figure 7 shows RF model estimates vs. observed snow depth.

Thanks. We corrected this in the revised manuscript (L426).

25. L424: 'more spatially continuous'. What do you mean with this? It seems to contradict the previous sentence.

We meant a continuous snowpack over the distance of the scale break. For example, the snowpack in Montmorency is more variable (high semi-variance value) and comparatively continuous over the distance of the scale break than the snowpack at the other two sites. (L469)

26. L444: I think here you should cite Mendoza et al. (2020b) and NOT Mendoza et al. (2020a).

Thanks. We cited this in the revised manuscript.

27. I think section 4.3 could be largely condensed. You may also consider shortening the introduction.

Thanks. We now divided this section to sub sections in the revised manuscript.

28. L479: 'At the combined scale'. I think it is more appropriate to write 'At the full domain'.

Thanks. We changed this in revised manuscript (L551).

29. L489: I think that you mean Hydrologic Response Units (HRUs).

Thanks. Yes. We corrected this mistake in revised manuscript (L562).

30. L490: 'successfully modeled'. Can you please provide some numbers demonstrating that the hydrologic modeling was indeed successful?

They used CRHM to model hydrological processes in agro-forested catchment in southern Quebec. They reported a NSE of 0.57 over 23-year simulation of SWE.

We changed this sentence as, "Aygün et al., (2020) successfully modelled (Nash–Sutcliffe efficiency of 0.57 over 23-year simulation of SWE) blowing snow transport in fields and the preferential accumulation in canals and streams, and assumed that once these were filled, any further blown snow accumulated in the forest" (L563-L566).

31.   L518: These results seem quite poor. Did you compare RF performance with multiple linear regression models?

Yes, we did. Performance of multiple linear models were poorer than the RF performance. We now added a comparison of them in the revised version (L634-L646).

32.   L532-533: I would not even mention those references, since NSE is not a good metric to assess the spatial accuracy of model simulations. There are other performance measures for such purpose (Koch et al. 2018; Demirel et al. 2018; Dembélé et al. 2020).

Thanks. They are examples of studies that used RF model to predict snow depths and SWE. They reported NSE as an accuracy measure. However, we added RMSE values they reported too to this section now (L621).

33.   L558: I would avoid referring to 'improved accuracy', since RF model results are quite poor.

Still, the inclusion of forest edge metrics does improve the accuracy, even if the overall resulting accuracy remains moderate. We adjusted the wording to "increased" in the revised manuscript (L670-L671).

34.   L567: 'forest structure variability'. Do you mean spatial or temporal variability?

We meant the spatial variability of the forest structure. We corrected this in revised version (L684).

**Suggested edits**

I have provided some editorial suggestions. However, I think that the manuscript would tremendously benefit from a language revision.

35.   L13: 'for the accurate prediction' -> 'for accurate predictions'.

Thanks. Changed as suggested.

36.   L28: delete 'problematic'.

Thanks. Changed as suggested.

37.   L34: 'on the downstream hydrograph' -> 'on downstream hydrographs'.

Thanks. Changed as suggested.

38.   L56: 'The knowledge' -> 'the estimation'.

Thanks. Changed as suggested.

39.   L60: delete 'modeling approaches like'.

Thanks. Changed as suggested.

40.   L77: 'that used RF algorithm to express' -> 'quantifying'.

Thanks. Changed as suggested.

41. Table 1: 'mm/y' -> 'mm/yr'.

Thanks. Changed as suggested.

42. L150: 'were quantified' -> 'were obtained'.

Thanks. Changed as suggested.

43. L153-155: I suggest writing the sentence between parentheses in a separate sentence.

Thanks. We added this information as a separate sentence now, "The manual sampling strategy consisted of five snow depth measurements taken at each sampling location in a diagonal cross shape at 1 m apart, and the average of these five measurements represent a 1.4x1.4 m ($\sqrt{1^2 + 1^2}$) grid cell."

44. L162: 'When taking into account' -> 'Considering'. Delete 'that was typically'.

Thanks. Considering referee #2 comments, we removed the L162–L168 section.

L45. L163: add a comma after 'environment'.

Thanks. Considering referee #2 comments, we removed the L162–L168 section.

46. L165: 'to represent' -> 'represent'.

Thanks. Considering referee #2 comments, we removed the L162–L168 section.

47. L165: 'As well': I find this term quite odd. I would replace by 'Additionally', 'Further', 'Moreover', etc. (this comment applies for the entire manuscript).

Thanks. We replaced the word "as well" with the suggested words throughout the manuscript.

48. L180: 'which gives' -> ', providing'.

Thanks. Changed as suggested.

49. L182: 'from the hourly' -> 'from hourly'.

Thanks. Changed as suggested.

50. L250: 'two thirds… is' -> 'two thirds… are'.

Thanks. Changed as suggested.

51. L266: delete 'of a variable'.

52.   L314: delete 'to allow'.

Thanks. Changed as suggested.

53.   L321-322: rewrite as '…LAI and WFE have the highest (64 %) and least (3 %) impacts, respectively, on snow depth…'

Thanks. Changed as suggested.

54.   L323: 'acting in forests and fields' -> 'in such environments'.

Thanks. Changed as suggested.

55.   L407: 'In our results' -> 'our results show that'.

Thanks. Changed as suggested.

56.   L415: '… suggests microtopographic… ' -> '…suggests that microtopographic…'

Thanks. Changed as suggested.

57.   L458-459: Awkward sentence. Please re-write.

We removed that sentence.

58.   L462: 'by the preferential' -> 'by preferential'.

Thanks. Changed as suggested.

59.   L470: Delete 'Whereas'.

Thanks. Changed as suggested.

60.   L475: 'and dominates' -> 'dominating'.

Thanks. Changed as suggested.

61.   L487: 'in the melting' -> 'during the melting'.

Thanks. Changed as suggested.

62.   L488: 'challenge the'. Delete 'the'.

Thanks. Changed as suggested.

63.   L512: 'could be due to' -> 'could be explained by'.

Thanks. Changed as suggested.

**Response to Anonymous Referee #2**

We thank the anonymous referee #2 for his/her thorough review with constructive comments and suggestions that certainly will improve the manuscript. In the following, we will address the referees' comments point by point. We mark "black" the comments given by the referee, and our responses in "blue". We also included the line numbers in revised manuscript in our responses.

**Comment on tc-2022-124**

Anonymous Referee #2

Referee comment on "Topographic and vegetation controls of the spatial distribution of snow depth in agro-forested environments by UAV-lidar" by Vasana Dharmadasa et al., The Cryosphere Discuss., https://doi.org/10.5194/tc-2022-124-RC2, 2022

The study by Dharmadasa et al. explores snow depth distribution at three Canadian sites (agro-forested and boreal forest) based on snow depth maps derived from UAV-LiDAR. Scaling behavior is investigated, and random forest models are used to assess the importance of various topographic and vegetation controls on these snow depth distributions. The topic of the study is of interest to the community, and in general, the manuscript is neatly organized, and the figures are well made. However, I have some methodological concerns that need to be addressed before this manuscript can be considered for publication. A major issue is certainly that the data at hand may not be sufficient to reach the conclusions drawn in the study – in the current state, the key findings and novelties of the study as well as the potential impacts of these findings are not highlighted well enough. Please find my major and minor comments detailed below, as well as some suggestions that I believe would make the study more novel and convincing.

We thank referee #2 for the challenging remarks and appreciate the overall potential he/she sees in our study. We acknowledge that the key findings, potential impacts of the findings and novelties were not highlighted well enough, and we addressed this thoroughly in the revised manuscript. We also considered the reviewer's suggestions in the revised version that will significantly improve the quality of the manuscript.

**Major comments:**

1. In the methods, the authors should state more explicitly that the datasets are published already. Some redundancy with their article published earlier this year is unavoidable but should be kept to a limit. There are instances where the text could be shortened and simply refer to Dharmadasa et al. 2022, I have pointed these out in the minor comments. Watch out for self-plagiarism - Figure 1 and Table 1 are almost identical to Dharmadasa et al. 2022, and need to include a proper reference (e.g. 'adapted from').

   Thanks. We added "adapted from" for Figure 1 and Table 1 and changed the text accordingly when we refer Dharmadasa et al. (2022).

2. I am not convinced by the choice of aggregating the variables to different resolutions (10-20m) for the RF modelling and related analysis, in my opinion this raises some problematic questions:

- Firstly, I don't understand why the aggregation of the vegetation parameters is needed at all (L200ff). 'Canopy height' and 'Tree height' is not necessarily the same thing, and I see no problem with computing canopy height if the pixel size is smaller than the tree crown. Doing so would actually allow extracting the small canopy gaps that have been shown to be the main source of forest snow variability in some other studies (cited in this manuscript), while these gaps are averaged out if the pixels are aggregated to 10-20m resolution. This averaging is likely masking some dependency of snow depth distribution on forest structure.

- Likewise, averaging topographic variables will average out most of the micro-topographic variability that I understood to be the focus of the study. Again, some dependency between snow depth and these variables may be masked by this aggregation. It should be clarified whether the authors are trying to quantify micro-topography or topography.

- I find it particularly problematic to use an aggregation that is larger than the scale break identified in Section 3.2. Doesn't this mean that the variability that you are trying to explain is averaged out?

- Finally, aggregating leaves you with a rather small sample size, as the surveyed areas are quite small. Especially in the case of Montmorency, the field landcover type covers a very limited area only.

- I would find the analysis more convincing if it was conducted at the 1.4 m resolution, I suggest doing that in view of a resubmission.

  The rational for the initial analysis was to study the snow cover heterogeneity at the scale of single-tree. But we agree with your comment that aggregating does average out the variability smaller than the single-tree scale we considered. Considering your comments and suggestion, we repeated the analysis at the 1.4 m resolution. Results show an overall improvement of the RF model performances at all sites (field+forest $R^2$ in Sainte-Marthe is 0.66, Saint-Maurice is 0.46 and Montmorency is 0.3 now). The importance of the variables at each site were only slightly changed. At agro-forested sites, windward forest edge effect and microtopography still have the highest impact on snow depth variability and in coniferous site, it is still the forest structure variability (LAI). But windward forest edge effect in forested areas at the two agro-forested sites are now evident compared to the coarse scaled data. We now replaced the whole RF analysis by the updated analysis done at 1.4 m resolution results in revised manuscript.

3. Some parts of the results chapters require more detail / explanation, and the discussion needs to be more convincing. Some examples:

- Section 3.1: The large overlap of forest and field histogram makes me wonder whether the difference between the two distributions is statistically significant – testing this would be appropriate (see e.g. Currier et al. 2018 for examples of such tests). The inter-site comparison is problematic because data acquisition did occur in different years.

  We tested the statistical significance of field and forest snow depths and included them in the revised version (L318-L322). Initial plan for the study was to acquire data in Montmorency in 2020 as well. But in compliances with the COVID-19 regulations that were in effect during that time, we were not allowed to access the site for surveys. However, the long term ECCC data shows that Montmorency always receives much higher snowfall than the two agro-forested sites. Also, the study years are representative of the long-term climatological conditions at the sites (Supplement Fig. S1). Therefore, we decided to utilize the data available at hand for the analysis.

- Section 3.2: It is unclear how the scale breaks were identified.

  We updated this information in 2.3.1 Spatial correlation analysis in the revised manuscript (L261-L269) based on extensive comments made by reviewer #1 on this point.

- Section 3.3.1: You need to better justify why you computed so many topographic and vegetation variables if most of these remain unused in the analysis / RF model. You also need to explain why you

used the same predictors at all sites – for instance, NFE and WFE are homogeneous across the forested area in Montmorency forest so it is not surprising that they have no predictive power.

We now better explained this in revised manuscript (L361-L381)

- Section 3.3.3: This section is a bit lengthy, and the key messages don't quite come through. Maybe it would be better to show fewer subpanels in Figure 6 and focus on the variables that actually exhibit interesting relationships to snow depth.

Thanks. We revised this section in revised manuscript (section 3.3.3). However, we decided to keep all sub panels since we also wanted to highlight that some variables (Aspect, NFE) do not have significant influence at 1.4 m grid scale. Yet, some of them (Aspect) can have a significant role in shaping snow accumulation and distribution patterns at scales larger than scale break distances found in this study (section 4.3.3).

- In the discussion, you need to comment on the actual benefit of such an RF approach. The model performance shown in 3.3.4 is pretty poor, and it's not straightforward to extract the interesting information from the plots in Figure 6. In fact, I felt like the most insightful Figure to grasp the snow depth variability was Fig 3, where increased accumulation in sheltered locations is visible by eye.

RF notably allows capturing non-linear relationships between snow accumulation and landscape variables. We now discuss the benefits of using RF approach in the updated discussion section, notably compared to traditional multiple linear regression models (section 4.4.3).

4. Given that the datasets have already been presented and used in an earlier study by the same work, the added value of the analysis presented in this manuscript seems a bit limited and the novelty of the study is not emphasized enough. What are the key findings, and where will they be useful? I realize that it's not easy to get more out of such a limited amount of data, but I tried to make sume suggestions that the authors could consider in view of a re-submission.

The earlier study solely focused on the accuracy of the UAV-LIDAR system, a necessary methodological step, and did not analyze the spatial heterogeneity of the derived snow depth maps. We thank the referee #2 for his/her insightful suggestions to improve the quality of our manuscript. We adopt the suggestions in the revised manuscript as much as possible.

- If the authors have the opportunity to acquire more data in the upcoming winter, the authors should consider postponing the final analysis to after the upcoming season. Repeated flights over the same site would allow for a much more insightful analysis. It is very difficult to draw conclusions on individual processes based on snow distribution data from one acquisition only, which I think is part of the reason why the discussion does not seem very conclusive. For instance, it could be interesting to see if the snow depth maxima at the forest edges are a recurring feature, and if their effect persists throughout ablation (that's just an idea, but one could do much more). It would also be good to survey all sites in the same year to allow a more convincing comparison between sites.

We agree that having repeated flights in the same areas in several winters would be better. But unfortunately, due to logistical reasons, we will not have the opportunity to acquire more data in the upcoming winter in these areas. This caveat is mentioned in the discussion. However, a new, ongoing study will explore the temporal variability of the snow depth in a coniferous site. Still the analysis presented here is thought to largely reflect the typical conditions at the sites and to portray key differences between agro-forested environments and boreal forests. To place the studied years in a climatological

context, we compared the climatological conditions in the study year (snowfall, total precipitation, air temperature, wind speed, and wind direction) with historical conditions at the sites and included them in the revised version (Supplement Fig. S1).

- Since the authors analyzed many more terrain and vegetation variables than they ended up using in the RF model, it could be interesting to dedicate a section on the physiographic variables themselves, attempting to identify a set of variables useful to characterize this sort of landscape. Maybe in comparison to variables that have been related to snow distribution in other studies. For example: Elevation has been found to exert a main control on snow depth in complex terrain in other studies, but it is not a really 'useful' predictor at the sites used in this study – is this a consequence of the site choice, or is this site representative of the terrain found in the whole ecoregion?

  We now added a section in the discussion in the revised version (section 4.5) to discuss the potential predictors at similar landscapes.

- Exploring the relationships between snow and physiographic variables at different spatial aggregation levels could be interesting.

  Thanks. The updated manuscript replaced the initial aggregated scale by the high-resolution 1.4 m analysis. We however included a section in the revised paper discussion to compare the RF results at 1.4 m scale to single-tree scale results (coarse scale we used earlier) (section 4.3.3).

- Applying additional modelling approaches (statistical, or even physically based) to compare with the RF model could be insightful, especially to draw conclusion on the utility of these findings for later work or practical applications.

  Thanks. We compared the RF results with multiple linear regression at 1.4m scale and added them in the revised version (section 4.4.3).

- Adding an application of the model results would be a nice addition – e.g. suggest tiling approaches, extend to larger area or entire watershed, etc.

  Thanks. We agree that this will be a great addition to our study. However, after including new sections and plots revised paper is now at 34 pages. As much as we appreciate this suggestion, we think the addition of this suggestion would make the revised paper lengthier. Moreover, we think we were able to highlight the novelty of the study by implementing suggestions 2,3, and 4.

**Minor comments** (including wording/language suggestions)

L37 'topography and vegetation type, and density' -> you mean vegetation density? Sentence doesn't read very well, consider rephrasing

Yes. It is vegetation density. We rephrased this sentence (L37).

L46: a very nice and comprehensive paper on the topic: https://doi.org/10.1029/2011WR010745 - I suggest including this reference

Thanks. We included this reference in the revised manuscript now (L46).

L51 'a short scale break is reflected by interception' -> I would say it's the other way round?

We meant that interception is causing the short scale break. We changed the sentence as "For instance, these studies emphasized that canopy interception causes a short scale break distance in forested areas (9–12 m) where the effect of wind redistribution is minimal" (L51-L53).

L62: You should refer to much more recent developments of process-based models, as some of these models now actually do resolve small-scale variability due to heterogeneous canopy structure. See Broxton et al. 2015 (already cited elsewhere) and Mazzotti et al. (https://doi.org/10.1029/2019WR026129 and https://doi.org/10.1029/2020WR027572).

Thanks. We now added some recent developments of process-based models (L63).

L93: 'one of the earliest results […]'. I think this is not a very fair 'selling argument' for this study, since the data used for the analysis has already been presented in another paper.

Thanks. We removed this sentence.

L108: incomplete sentence (WMO's station network?)

Thanks. It is WMO's station network. We corrected this (L111).

Table 1: Winter season -> snow cover period?

Yes.

L136 is this vertical or horizontal accuracy, or both?

It is vertical accuracy. We added this information now (L142).

L164: what do you mean by 'multipath effect'?

Internal reflection of GNSS signals against obstacles (trees) before reaching the receiver.

L165: you just said the accuracy is comparable to previous studies, so what is the improvement? I would omit the entire end of the paragraph from 162 onward and just refer to Dharmadasa 2022.

We removed the section from L162 onwards and referred to Dharmadasa et al. (2022).

Section 2.2.2-2.2.4: Please specify that maps of the variables are found in the supplementary material, I was missing those maps here and found them only much later. Note that the figures in the supplement should include the units for all variables.

Thanks. We corrected this in the relevant sections and added the units in supplement figures.

L197-199 Is GC = 1-CC? and at what resolution are these metrics calculated, also 1.4m?

Yes. They were also calculated as the same size as the LAI. We added this information in the revised version (L185-L198).

L201: How did you estimate crown diameter?

We calculated the crown diameter in LiDAR360 software. LiDAR360 uses Li et al. (2012) point cloud segmentation algorithm to segment individual trees. Crown diameter is one of the outputs we get from these segmented trees in the software (Greenvalley-International, 2020). We added more details in the revised version now (L296-L302).

L224: This approach seems a combination of Currier & Lundquist and the DCE presented by Mazzotti et al 2019 (which however has no notion of search distance contrary to your method). Maybe worth noting?

Thanks. We now included this in the revised version (L207).

L256: It is not very clear how you define the variable 'Site', or at least it wasn't to me when looking at the descriptor maps in the supplementary material and comparing with the other vegetation metrics. I think this is quite crucial for understanding the edge metrics, hence I more detail is needed here.

The binary variable, site was derived according to the field and forest area boundaries manually mapped at each study area. If there is a forest patch in the field, we considered that patch as forest and vice versa. For instance, in Sainte-Marthe we considered the forest patch located to the southwest in the field as forest in addition to the large forested area. But in Saint-Maurice and Montmorency, derivation of site variable was more straightforward as field and forest patches were well separated and so easily distinguishable than in Sainte-Marthe (i.e., there were no forest patches in field as in Sainte-Marthe). Once we delineate the forest and field boundaries, we assign 0 to field and 1 to forest.

We now added a separate section "2.2.4. Site variable" to explain this better.

L264 Tenses are inconsistent

Thanks. Corrected in the revised version.

L266: Variable importance of a variable? Consider rephrasing

Thanks. We removed the whole sentence when rephrasing this section. However, when we refer to this, we now used variable importance or importance of a variable.

Figure 3: specify whether you used the binary variable or the land cover classification to create the histograms.

We used the boundaries from site variable to create the histograms. Figure 3 was (now Figure 5) updated in the revised manuscript indicating the forest and field boundaries.

L308: how did you come to this conclusion?

Depending on the correlation coefficient value. Please note that in the updated analysis at 1.4 m grid resolution, we now included the leeward forest edge (LFE) variable in Montmorency RF model, since it has more influence on the snow depth predictions than windward forest edge variable (which was also confirmed from the larger extent of LFE raster than WFE). We rephrased this whole section now (section 3.3.1).

L310 'collinearity analysis suggested discarding GF and CH in favor of LAI at the two agro-forested sites, while LAI was instead flagged as colinear instead of GF and CH in the coniferous site'. Please rephrase – 'LAI was flagged as colinear' is unclear (colinear to what?)

Thanks. We rephrased this whole section in the revised version (section 3.3.1).

Figure 6: Y-axis label missing (snow depth)

Thanks. We corrected this in the revised version (now Figure 8).

L445-446: Unloading through branches should reduce spatial variability, no?

Yes. It could. Lower semi-variance value in temperate forests compared to coniferous forest in figure 5 suggests that the overall spatial variability of the snowpack is less in these forests compared to that in coniferous forest. Yet, this could still result in a smaller correlation length in snow depth because of the interwind nature of branches on deciduous trees.

L511-513: The counterintuitive […] stations at the site. -> I don't understand what you are trying to say here.

We wanted to say, in contrast to preferential deposition on northerly slopes in the northern hemisphere, Montmorency field accumulates more snow on southerly slopes. This could be due to the influence of the various meteorological stations at the site. We removed this sentence in the revised version as at 1.4 m resolution, no apparent snow accumulation is observed on northerly or southerly slopes.

L565ff: this section needs to acknlowledge hyper resolution process based (physically based) models (see earlier comment). There are ways to account for fine scale canopy structure, while I would say that the terrain roughness still represents a major difficulty.

Thanks. We addressed this in the revised version (L682-L683).

L483: I think this should be 'Hydrologic response units'

Yes. Thanks. We corrected this in the revised version (L562).

Section 4.3: The work from Safa et al. (https://doi.org/10.1029/2020WR027522) needs to be included in this discussion – they applied RF models as well.

Thanks. We added this reference in the revised version (L622).

**Response to Anonymous Referee #3**

We thank the anonymous referee #3 for his/her thorough review with constructive comments and suggestions that certainly will improve the manuscript. In the following, we will address the referees' comments point by point. We mark "black" the comments given by the referee, and our responses in "blue". We also included the line numbers in revised manuscript in our responses.

Comment on tc-2022-124 Anonymous Referee #3 Referee comment on "Topographic and vegetation controls of the spatial distribution of snow depth in agro-forested environments by UAV-lidar" by Vasana Dharmadasa et al., The Cryosphere Discuss., https://doi.org/10.5194/tc-2022-124-RC3, 2022

**Major comments:**

1. This manuscript is using UAV-lidar data to understand the snow-depth heterogeneity in agro-forested environments and boreal forests. Since the author also used the slope and aspect as topographical variables for studying the snow-depth distribution. However, at site Saint-Marthe and Saint-Maurice, the elevation difference is actually very small comparing to the spatial scale of the study area, it would be kind of surprising to observe any meaning effects from topographical variables in these 2 study areas. And although Montmorency has a elevation difference of 20 meters in the study area, it seems the main area is facing towards south east direction. It would also be helpful to visualize the distribution of the slope and aspect of the 3 study areas to make sure there is significant variability in these predictors of snow-depth before feeding them into a model like random forest.
   Thanks. Please kindly note that we included the plots of the independent variables in the supplementary material (Supplement Figure S2-S4).

2. It is not very clear on how the forest edge descriptors are derived, it would be helpful to have a visualized illustration to demonstrate how variables are derived based on canopy cover data from lidar.
   Thank you for the suggestion. We included a visualized illustration (Figure 3) to demonstrate how forest edge descriptors were derived in the revised manuscript.

3. The 2nd objective raised in the Introduction part for this manuscript seems to be extremely open ended. Is there a particular hypothesis the authors would like to test with the used dataset and validate the hypothesis throughout the manuscript? The current objective of "exploring the relationship between snow depth, topography, and forest structure" seems too vague and not specific enough.
   Thank you for pointing this out. We made the second objective more specific in the revised manuscript. We added the following sentences at the end of this paragraph. "Motivated by previous works (Currier and Lundquist, 2018; Mazzotti et al., 2019), we specifically investigate how the forest edges modulates the accumulation patterns in agro-forested environments. Given the relatively flat topography in these environments, we postulate that preferential accumulation along forest edges may represent a significant factor of spatial variability in snow depth" (L95-L98).

**Minor comments;**

Figure 1 – it looks like the first 2 study areas are very flat with low elevation and the 3rd one has elevation difference and the elevation is much higher. It the precipitation in this area affected by orographic effect as well?
At the scale we considered for our study, elevation difference at any of the sites were too small to produce any meaningful orographic effects. Even at larger scales, the first two sites are generally flat, hence will not have any orographic effects on precipitation. But in the third site, Montmorency, when the scale of the study

area becomes larger (beyond that analyzed here), orographic effects would be expected to have an impact on precipitation.

Line 154-155: given it is 1 m diagonal cross shape, why it is 1.4x1.4 m grid cell? Isn't it going to be 0.7m x 0.7m grid cell instead?
The manual measurement strategy with 5 measurements at each sampling location we adapted was as follows:

[Figure]

Hence, the length in one side of the grid cell is $(1^2+1^2)^{1/2}$. Which is 1.4m.

Line 162-165: it is not clear why UAV-lidar is more robust and the technology represents an improvement to previous studies.
By considering your comment and a similar comment from referee #1 and #2, we removed section L162–L168 and referred to Dharmadasa et al. (2022).

Line 184-186: The closest weather station is 19 km away from Saint-Maurice. Is the wind data going to be trustable for this site given it is very far and the wind speed and direction can be quite different comparing to the actual on the site, right?
This is the closest wind station available for this site. Given the flat topography of this area, wind speed is expected to be not driven by topography and is spatially coherent over large scales.

Line 193: how is LAI, CC, and GF calculated? By using Lidar360 software?
Yes. They were calculated in Lidar360 software. This now is better explained in the revised version (L185-L198).

Line 202: why the grid size for vegetation is so much larger than the resolution of the snowdepth (1.4x1.4 m). It seems the vegetation grid resoluation is so much coarser and are we able to capture all the forest variable based on such a low resolution?
The initial goal was to analyze snow depth variability at a scale of single-tree level. We selected the resolutions in a way that at least one tree will be inside the grid cell. However, we agree that the variability averages out inside a coarse grid cell. Based on your comment and that of referee #2 we redid all analyses at the 1.4 m resolution. Random forest analysis results and discussion sections were replaced accordingly in the revised manuscript.

Section 2.2.4, please see major comment #2, it is not very clear how d and d_max are derived based on the forest-covered lidar data.
We added an illustration to better explain the derivation of d and d_max.

Line 250-253, are hyperparameters in Random Forest tuned or selected before training each model?

Hyperparameters in random forest at each site were tuned before training each RF model. We now added this in the revised version (L285-L286).

Line 257: It is not very clear how forested vs. fields are defined. It would be helpful to have a map of these site showing this binary variable. And why don't we use this binary variable directly in the RF model directly? Is it to show at different area how other variables affecting snow-depth differently?

The binary site variable was derived according to the field and forest area boundaries manually mapped at each study area. If there was a forest patch in the field, we considered that patch as forest and vice versa. For instance, in Sainte-Marthe we considered forest patch located to the southwest in the field as forest in addition to the large forested area. In Saint-Maurice and Montmorency the derivation of the site variable was more straightforward as field and forest patches were well separated and so more distinguishable than in Sainte-Marthe (i.e., there were no forest patches in field as in Sainte-Marthe). Once we delineate the forest and field boundary, we assign 0 to field and 1 to forest.

We now added a separate section "2.2.4. Site variable" to explain this better.

Please note that the plot of site variable at each site are included in the supplement (Supplement Figure S2-S4).

Figure 3: it is a bit surprised to me at Montmorency there is not many data points for under canopy. Then we might not be able to observe a lot of under canopy snow-depth signals.

Since Montmorency has a thick evergreen canopy cover, we did not get ground returns under the canopy at some locations. Hence, we lacked the snow depths under canopy on such occasions. This which is a limitation of the UAV-LIDAR system in thick coniferous covers is acknowledged in section 4.4 Limitations of the study.

Figure 4: how is the scale break selected? Please describe that in the Method section.

We updated this information in 2.3.1 Spatial correlation analysis in the revised manuscript based on extensive comments made by reviewer #1 on this point (L261-L269).

Figure 5: it might be better to use bar chart with different colors. It is a bit difficult to differentiate color the marker styles on this scatter plot.

Thanks. By considering your comment and a similar comment from referee #1, we linked symbols in figure 5 (now Figure 7) with straight lines in the revised manuscript for a clear interpretation.

Section 3.3.3: there is only one line in the Method section (line 267) discussed about the partial relationships of preditor variables with snow depth. It is still not very how that is calculated. Please add details in the Method section.

Thanks. We added more details about how partial relationships were derived in the revised version (L288-L293).

Figure 7, it looks like the model is not performing very well at Saint-Maurice and Montmorency. The slope of the predictioned vs. observed is not close to 1. What would be the reason that the trained RF model is underfitting and has this systematic bias?

The performance improved with the higher (1.4 m) resolution model (now Figure 9), but the poorer performance at Saint-Maurice and Montmorency could be due to underlying processes/variables that were not considered in our model, as well as unexplained snow depth variability that is within the system (UAV-LIDAR) detection limit. We added this issue to the updated discussion (L607-L617, L658-L664).

Currier, W. R. and Lundquist, J. D.: Snow depth variability at the forest edge in multiple climates in the western United States, Water Resour. Res, 54, 8756–8773, doi: 10.1029/2018WR022553, 2018.

Deems, J. S., Painter, T. H., and Finnegan, D. C.: Lidar measurement of snow depth: a review, Journal of Glaciology, 59, 467–479, doi: 10.3189/2013JoG12J154, 2013.

Dharmadasa, V., Kinnard, C., and Baraër, M.: An accuracy assessment of snow depth measurements in agro-forested environments by UAV lidar, Remote Sensing, 14, 1649, doi: 10.3390/rs14071649, 2022.

GreenValley-International: LiDAR360 User Guide, GreenValley International, Ltd, Berkeley, CA, USA, 2020.

Greenwell, B. M.: pdp: An R package for constructing partial dependence plots, The R Journal, 9:1, 421–436, 2017.

Harder, P., Pomeroy, J., and Helgason, W.: Improving sub-canopy snow depth mapping with unmanned aerial vehicles: Lidar versus structure-from-motion techniques, The Cryosphere, 14, 1919–1935, doi: 10.5194/tc-14-1919-2020, 2020.

James, N. A. and Matteson, D. S.: ecp: An R package for nonparametric multiple change point analysis of multivariate data, Journal of Statistical Software, 62, 1–25, doi: 10.18637/jss.v062.i07, 2014.

Jerome, H. F.: Greedy function approximation: A gradient boosting machine, The Annals of Statistics, 29, 1189–1232, doi: 10.1214/aos/1013203451, 2001.

Koutantou, K., Mazzotti, G., and Brunner, P.: UAV-based lidar high-resolution snow depth mapping in the swiss alps: Comparing flat and steep forests, Int. Arch. Photogramm. Remote Sens. Spatial Inf. Sci., XLIII-B3-2021, 477–484, doi: 10.5194/isprs-archives-XLIII-B3-2021-477-2021, 2021.

Li, W., Guo, Q., Jakubowski, M. K., and Kelly, M.: A new method for segmenting individual trees from the lidar point cloud, Photogrammetric Engineering & Remote Sensing, 78(1), 75–84, 2012.

Liaw, A. and Wiener, M.: Classification and Regression by randomForest, R News, 2/3, 18–22, 2002.

Mazzotti, G., Currier, W., Deems, J. S., Pflug, J. M., Lundquist, J. D., and Jonas, T.: Revisiting snow cover variability and canopy structure within forest stands: Insights from airborne lidar data, Water Resour. Res, 55, 6198–6216, doi: 10.1029/2019WR024898, 2019.

Mendoza, P. A., Musselman, K. N., Revuelto, J., Deems, J. S., López-Moreno, J. I., and McPhee, J.: Interannual and seasonal variability of snow depth scaling behavior in a subalpine catchment, Water Resour. Res, 56, e2020WR027343, doi: 10.1029/2020WR027343, 2020.

Probst, P. and Boulesteix, A.-L.: To tune or not to tune the number of trees in random forest, Journal of Machine Learning Research 18, 6673–6690, 2017.

Sun, W., Xu, G., Gong, P., and Liang, S.: Fractal analysis of remotely sensed images: A review of methods and applications, International Journal of Remote Sensing, 27, 4963–4990, doi: 10.1080/01431160600676695, 2006.

Tyralis, H., Papacharalampous, G., and Langousis, A.: A brief review of random forests for water scientists and practitioners and their recent history in water resources, Water, 11, 910, 2019.

---

## Author Response (AR2)

**Response to Anonymous Referee #1**
We thank the anonymous referee #1 for reviewing the manuscript for the second time, his/her comments, and suggestions. In the following, we will address the referees' comments point by point. We mark "black" the comments given by the referee, and our responses in "blue". We also included the line numbers in revised manuscript in our responses.

**Comment on tc-2022-124**
Anonymous Referee #1
I want to thank the authors for the detailed replies to all my comments, and I commend them for re-doing a lot of analyses and including new results. I think that the main concerns raised in the previous revision have been thoroughly addressed, and the manuscript reads much better. I have a final set of observations and editorial suggestions that will help the authors to put this paper in publishable format.

Minor comments

1. Section 2.2.2: can you please clarify how does the TWSI parameter relate to the TPI or the Sx parameter in Revuelto et al. (2014)?

We added a sentence addressing this in section 2.2.2 now (L173–L174).

2. Section 2.3.1: please clarify what you mean with 'topography+vegetation'. Bare earth topography + forest trees?

Thanks. We added this information in L247 now.

3. I think that subsection 3.3.1 should be in the Methods section, and NOT in the results section.

Thanks. We now moved this section to section 2.3.2 Random forest model in methods (L288–L307).

4. L495 (section 4.2.1): are bare earth topography variograms actually comparable to snow depth field variograms? Did you use exactly the same domains to compute these? Please clarify.

Yes. We used bare earth topography in the field to compute the variogram. We now mentioned this in L246–L248.

Suggested edits

5. Abstract: I suggest adding a connector between the first two sentences.

Thanks. We modified the sentence as suggested now.

6. L19: large -> larger.

Thanks. Changed as suggested.

7. L22: add a comma after 'Hence'.

Thanks. Changed as suggested.

8. The last sentence of the abstract seems redundant. I suggest deleting it.

Thanks. Changed as suggested.

9. L31: 'melt dynamics of the snowpack' -> 'snowmelt dynamics'.

Thanks. Changed as suggested.

10. L60: 'aspect in understanding' -> 'aspect for understanding'.

Thanks. Changed as suggested.

11. L72-74: 'the random forest (RF) model… started gaining popularity' -> 'random forests (RF) models …. have gained popularity'.

Thanks. Changed as suggested.

12. I personally dislike the use of the word 'successfully' or 'successful' in scientific writing, without providing. If you want to use them, I think that, at the very least, you should show the numbers that make a method 'successful'.

Thanks. Changed as suggested.

13. L97: postulate -> hypothesize.

Thanks. Changed as suggested.

14. L102: delete 'of the three sites'.

Thanks. Changed as suggested.

15. L113: 'for the interpretation purpose' -> 'for interpretation purposes'.

Thanks. Changed as suggested.

16. L405: there is a typo here (Aspest_SN).

Thanks. Corrected.

17. L585: create -> creating.

Thanks. Changed as suggested.

18. L652: '…and land cover' -> '…, land cover' (i.e., replace 'and' by comma).

Thanks. Changed as suggested.

19. L672: 'process-based models' -> 'physically-based models'.

Thanks. Changed as suggested.

20. L679: delete 'most of the process-based'.

Thanks. Changed as suggested.

References

Revuelto, J., J. I. López-Moreno, C. Azorin-Molina, and S. M. Vicente-Serrano, 2014: Topographic control of snowpack distribution in a small catchment in the central Spanish Pyrenees: intra- and inter-annual persistence. Cryosph., 8, 1989–2006, doi:10.5194/tc-8-1989-2014.

**Response to Anonymous Referee #2**
We thank the anonymous referee #2 for reviewing the manuscript for the second time, his/her comments, and suggestions. In the following, we will address the referees' comments point by point. We mark "black" the comments given by the referee, and our responses in "blue". We also included the line numbers in revised manuscript in our responses.

**Comment on tc-2022-124**
Anonymous Referee #2
I am reviewing this manuscript for the second time, and I recognize and appreciate that substantial effort went in this revision to address the reviewer comments on the first version of the manuscript. This has clearly improved the manuscript; in particular, I find the analysis at higher spatial resolution more convincing than the original one.

There are a few remaining issues that need to be fixed before publication; except for one (see 'major comments'), they mainly relate to readability, clarity, and logic – I felt that some statements were contradictory. In general, I think manuscript readability would benefit from some shortening, which could be achieved by more precise and concise language and by removing some repetition in the discussion.

Please note that line numbers in my comments refer to the revised manuscript without tracked changes.

Major comments:

1. In L20ff, the authors state that 'Results show that […] increased the model prediction accuracy by more than 90 %.', and the same statement reappears at the beginning of the conclusion. I find this statement a bit problematic for two reasons: Firstly, without context, it is not clear what this 90% improvement is compared to, nor where the number comes from. Moreover, it is presented as a major result of the study in the abstract and the conclusion, while there are multiple instances where the authors state that predictive accuracy was not the primary concern of the study. This is contradictory and should be handled more consistently, I believe highlighting the other main findings would be more appropriate

You are correct that some clarifications were needed. Indeed, the main focus of the analysis was on identifying the drivers of variability and less on the prediction accuracy itself, although relevant and also discussed in the paper. This is why, for example, we carefully screened predictors for collinearity. Here, we do not compare the model accuracies, but instead the *increase in model accuracy* (decrease of prediction error) when these key predictors are considered. The relative importance of a variable shows how much the inclusion of the variable of interest decrease the mean squared error (MSE) of the RF model prediction *compared to a model that discards the variable of interest*. As such the large relative importance (given as %) of WFE in agro-forested sites and LAI in coniferous site imply that including these predictors improve the model prediction accuracy greatly compared to a model that discards these effects. We acknowledge that this was not clearly presented in the paper. We now modified the abstract, methods, discussion, and conclusion to better explain this.

2. L271ff: This paragraph is a bit confusing. The authors first state that data was not split, but then argue that the general procedure IS to split the data and explain the in-bag / out-of-bag concept. It remained unclear to me what exactly was done for this study (i.e. whether data was splitted or not).

We modified the text to clarify the procedure (L270–L280). We wanted to highlight that we did not keep a set-aside test set from our sample data. Instead, we used all the available data in RF model and used the OOB statistics for model evaluation. The RF model itself randomly splits the input data into in-bag and out-of-bag sets, train the model using the in-bag set and validate the model using the out-of-bag set.

Minor comments:

L44: 'high-resolution micro variability' -> I suggest dropping high-resolution to avoid repetition in the same sentence

Thanks. Changed as suggested.

L 54: 'while a shorter (6 m) and longer (20 m) distance in non-vegetated areas are explained' -> Should be changed to shorter distances (plural)

Thanks. Changed as suggested.

Section 2.2.4 I appreciate that the site variable is now better explained, but it took me a while to understand that the classification relied on the land cover map only; maybe it would be a good idea to refer back to Fig 1 here.

Thanks. Added as suggested (L199).

L234: The 'DCE' descriptor from Mazzotti et al. (2019) is also continuous, hence this statement is not strictly correct. I think the novelty is mainly that the edge metric used in this study takes wind direction into account. I suggest reframing this statement.

Thanks. We reframed this now (L231–L233).

L290: 'It is a risk-adjusted alternative to variable dependence.'. this statement (especially 'risk-adjusted') is unclear, consider rewording

Thanks. Changed as "better alternative" (L311).

L305ff: I am not convinced that listing the max. snow depth values at each site is very meaningful since the values all seem outliers at the upper end of the histogram. I would omit this for the sake of making this section more concise.

Thanks. We now removed the maximum values reported in the Figure 4 and the section.

L420: 'probably due to the influence of instrumentation': It is unclear what instrumentation this refers to.

Thanks. Changed the sentence as "influence of instrumentation at NEIGE-FM site" (L414).

L430 'than their forest models': consider rephrasing to 'than the forest model at the same site' (or 'than the corresponding forest model'), I believe this would be more precise.

Thanks. Changed as suggested (L424).

L445: 'The overall amount of snow in the forest compared to field in the boreal forest of Montmorency could thus be underestimated due to poor lidar coverage under dense canopies': It is unclear why a lower coverage implies an underestimation of snow depth

Snow depth actually has to be "overestimated". We apologize for this mistake. We now explain this better (L442–L445).

465: 'or larger-scale effects': It is unclear what this refers to

We corrected this as larger-scale topographic effects now (L455).

472: 'Shorter scale break distances in forested areas compared to open field areas (Fig. 5) is analogous to previous studies that studied the fractal distribution of snow depths with lidar data.' : This statement doesn't seem supported by the references listed in the following sentences, please revisit.

Thanks. We rephrased this section now (L471–L478).

L514ff: For modelling, this is only true if the goal of the modelling is to fully resolve the variability, but not strictly necessary – a sub-grid parametrization is also an option

Thanks. Considering your comment, we now changed the wording "represent" to "fully resolve" to better explain this (L513).

530ff: I suggest removing this introductory sentence to avoid redundancy

Thanks. We now removed this introductory part.

L581: 'microtopography has a more restricted influence on deeper snowpack at this site compared to the shallower snowpack at the agro-forested sites'; vs L591: 'Our findings however show that the microtopography, even under wind-sheltered conditions in the forest, still plays an important part of the spatial variability.' – Note that while not contradictory, contrasting statements like these confound the key message of the paragraph. I would strongly advise to streamline such discussion sections.

Thanks. We rephrased the latter part now (L588–L589).

Chapter 4.3.3 somewhat disrupts the story line and refers to results that are presented in the supplementary material only. I would recommend moving the entire section to the supplementary material.

Thanks for this suggestion. We decided to remove this section altogether from the paper now not to disrupt the flow.

Chapter 4.4.3: the table comparing RF and MLR performance seems better suited to the results section, I suggest reconsidering this

Thanks for this suggestion. We moved the table 3 to results section "3.3.4. Performance of RF models at each site".

616: 'grid-scale mean snow depth': It is unclear what this means

Thanks. We just use "grid-scale snow depth" now (L600).

Chapter 4.6: It would be nice to see statements on two additional aspects: 1. Amount and coverage of your data – how transferrable are insights obtained from so little data only? 2. The fact that statistical methods provide only limited understanding of the processes underlying the observed patterns.

Thanks. This has been addressed in section 4.6 now.

679: 'More often': Than what? Consider omitting.

Thanks. Changed as suggested.

691: 'would benefit from incorporating the meteorological conditions': It is unclear what 'meteorological conditions' refers to and which result allows drawing this conclusion, please specify.

That confusing statement has been removed altogether.